# Ampere-level CO$_2$ electroreduction with single-pass conversion exceeding 85% in acid over silver penetration electrodes

Shoujie Li[1,2,5], Xiao Dong [1,2,5], Gangfeng Wu[1,2,5], Yanfang Song[1,2], Jianing Mao[1,3], Aohui Chen[1,2,4], Chang Zhu[1,2], Guihua Li[1,2], Yiheng Wei[1,2], Xiaohu Liu[1,2,4], Jiangjiang Wang[1,2], Wei Chen [1,2] ✉ & Wei Wei [1,2,4] ✉

Synthesis of valuable chemicals from CO$_2$ electroreduction in acidic media is highly desirable to overcome carbonation. However, suppressing the hydrogen evolution reaction in such proton-rich environments remains a considerable challenge. The current study demonstrates the use of a hollow fiber silver penetration electrode with hierarchical micro/nanostructures to enable CO$_2$ reduction to CO in strong acids via balanced coordination of CO$_2$ and K$^+$/H$^+$ supplies. Correspondingly, a CO faradaic efficiency of 95% is achieved at a partial current density as high as 4.3 A/cm$^2$ in a pH = 1 solution of H$_2$SO$_4$ and KCl, sustaining 200 h of continuous electrolysis at a current density of 2 A/cm$^2$ with over 85% single-pass conversion of CO$_2$. The experimental results and density functional theory calculations suggest that the controllable CO$_2$ feeding induced by the hollow fiber penetration configuration primarily coordinate the CO$_2$/H$^+$ balance on Ag active sites in strong acids, favoring CO$_2$ activation and key intermediate *COOH formation, resulting in enhanced CO formation.

The electrochemical conversion of CO$_2$ driven by renewable electricity can produce value-added chemicals and feedstocks while mitigating CO$_2$ emissions[1–3]. Numerous efforts have been made to develop catalysts with high current density ($j > 1$ A/cm$^2$) and high faradaic efficiency (FE > 90%) toward CO, formate, etc[4–9]. Typically, alkaline or neutral electrolytes are used to suppress the competing hydrogen evolution reaction (HER) while promoting the electrocatalytic CO$_2$ reduction reaction (CO$_2$RR)[10–12]. However, during CO$_2$RR and HER, the rapid consumption of H$^+$ creates a locally alkaline environment close to the catalyst surface. Consequently, rather than being reduced, a major fraction of the input CO$_2$ is consumed in the electrolyte via reaction with hydroxide ion (OH$^-$) to produce (bi)carbonate[13–17]. In addition, transporting (bi)carbonate to the cathode flow field or anode results in

a significant reduction of locally available CO$_2$ and a low CO$_2$ single-pass carbon efficiency (SPCE), impeding the practical applications of CO$_2$ electrolysis[18–21].

One strategy for addressing these issues is conducting CO$_2$RR in an acidic medium[22–28]. That is, in a catholyte with a low pH, when the hydronium (H$_3$O$^+$) serves as the proton source for CO$_2$RR and HER, no hydroxide ion (OH$^-$) will be generated, and CO$_2$ conversion can proceed without (bi)carbonate formation; even when H$_2$O is the proton source, any OH$^-$ or (bi)carbonate generated locally will be neutralized or converted back to CO$_2$ by protons in the bulk electrolyte, preventing CO$_2$ from transferring to the anode[22,29]. However, efficient CO$_2$RR in an acidic medium is difficult due to the kinetically superior HER outcompeting the reduction of CO$_2$[23,30,31]. For instance, in a strong

[1]Low-Carbon Conversion Science and Engineering Center, Shanghai Advanced Research Institute, Chinese Academy of Sciences, Shanghai, China. [2]State Key Laboratory of Low Carbon Catalysis and Carbon Dioxide Utilization, Shanghai Advanced Research Institute, Chinese Academy of Sciences, Shanghai, China. [3]Shanghai Institute of Applied Physics, Chinese Academy of Sciences, Shanghai, China. [4]School of Physical Science and Technology, ShanghaiTech University, Shanghai, China. [5]These authors contributed equally: Shoujie Li, Xiao Dong, Gangfeng Wu. ✉e-mail: chenw@sari.ac.cn; weiwei@sari.ac.cn

acid with a pH ≤ 1, the FE of the $CO_2RR$ product is nearly close to zero[23]. One of the main reasons is that the adsorbed hydrogen (*H) acts as an intermediate for HER, out-competing the adsorption of $CO_2$ (*$CO_2$) overactive sites during $CO_2RR$ in an acidic medium[23,30,31]. Recently, it was discovered that $K^+$ in the electrolyte could shield the electrode electric field and inhibit the transport of hydrogen ions ($H^+$) and that the rapid consumption of surface $H^+$ at high $j$ could increase the pH near active sites, allowing for efficient $CO_2RR$[22–25]. Several electrocatalysts, including $Au$[24], $Ag$[32], $Cu$[24], and $Ni_5@NCN$[33], have demonstrated the ability to $CO_2RR$ in acid, but their high $CO_2RR$ selectivity ($FE_{CO2RR} > 80\%$) could only be achieved in a limited range of $j$ (≤0.5 A/cm$^2$) with a low rate of product formation, which hinders their scalable applications.

Recently, a hollow fiber penetration electrode (HPE) with a compact structure has shown promising potential for high-rate and efficient $CO_2$ reduction due to enhanced mass transport[34–39]. The unique three-dimensional electrode structure compels gaseous $CO_2$ to permeate its abundant pores; the adequate oriented mass transfer at extensive triphasic reaction interfaces significantly improves the electrocatalytic kinetics. Herein, an $Ag_2CO_3$-derived hierarchical micro/nanostructured silver HPE (CD-Ag HPE) was used to investigate the effects of catalyst microenvironments (such as local concentrations of $K^+/H^+$ and $CO_2$) on $CO_2$ electrolysis performance in an acidic medium (pH = 1) (Fig. 1). By optimizing catholyte composition ($H^+$, $K^+$ concentration) and input $CO_2$ flow rate, a high CO current density ($j_{CO}$) of 4.3 A/cm$^2$ with CO FE of 95% and stable electrolysis of 200 h at 2 A/cm$^2$ were achieved in a strongly acidic electrolyte. Furthermore, a $CO_2$ SPCE of over 85% was achieved at 2 A/cm$^2$ by modulating the availability of $CO_2$. In addition, the density functional theory (DFT) calculations indicated that the coexistence of $H^+$ and $K^+$ localized around the Ag sites played a crucial role in the formation of the key intermediates *COOH and *H, which not only suppressed the competitive HER but also promoted the $CO_2RR$.

## Results and discussion

### Electrode preparation and characterization

The CD-Ag HPE was fabricated via a two-step approach[37] that was based on commercial Ag powder; additionally, it included an industrially viable phase-inversion/sintering process to obtain Ag HPE and an electrochemical redox process to obtain CD-Ag HPE (Supplementary Fig. 1). Compared to Ag HPE, CD-Ag HPE was about 20 μm thick, with partially ordered nanorods evenly coating the outer surface (Fig. 2a–c and Supplementary Fig. 2). This unique hierarchical micro/nanostructured architecture provided an increased electrochemical active surface area (ECSA, Supplementary Fig. 3) that maximized the three-phase reaction interfaces and enabled the efficient transport of reactants and products to/from the active sites for high-efficiency electrocatalytic reaction[34–39]. The

X-ray diffraction (XRD) patterns were indexed to metallic Ag (111), (200), (220), (311), and (222) planes (JCPDS no. 04-0783), and there was no obvious crystal face orientation, which was almost the same as those of commercial Ag powder (Fig. 2d and Supplementary Figs. 4a, b). The X-ray photoelectron spectroscopy (XPS) confirmed that the surface compositions of CD-Ag HPE were identical to that of metallic silver (Fig. 2e and Supplementary Fig. 4c). In addition, the selected-area diffraction (SAED) pattern (inset of Fig. 2f) of CD-Ag HPE agreed well with the XRD results. Supplementary Fig. 5 depicted the high-resolution transmission electron microscopy (HRTEM) image and the corresponding fast Fourier transform (FFT) pattern of the marked region, in which only metallic Ag was observed.

### Electrocatalytic $CO_2RR$ in neutral electrolyte

The electrochemical experiment of the $CO_2RR$ was conducted in two chamber electrolysis cell with a three-electrode system at room temperature, where the CD-Ag HPE was used as the working electrode and gas diffuser (Fig. 2g–i). During $CO_2$ electroreduction, $CO_2$ penetrated through the porous wall of the CD-Ag HPE into the electrolytes via the copper tube, forming a large amount of bubbles. This unique oriented mass transfer of $CO_2$ could induce the in-situ formation of extensive dynamic $CO_2$(gas)–liquid–catalyst triphasic reaction interfaces, which significantly improve the mass transfer of $CO_2$, electrons, protons, products as well as $CO_2RR$ kinetics[34–39]. Subsequently, a mixture of CO produced by $CO_2RR$, $H_2$ produced by HER, and unreacted $CO_2$ flows out through an outlet connected to the top right of the electrolysis cell. The actual outlet flow rate was measured by an independent mass flowmeter and then sent to online gas chromatography (GC) for quantification. And the exhaust from the GC was vented to the outdoor hood (Fig. 2h, i). Unless otherwise specified, a flow rate of 30 standard cubic centimeters per minute (sccm) was used to compare our performance to that of other studies[34–36]. It was shown that only CO and $H_2$ were detected at the $j$ range of 0.1–4.0 A/cm$^2$ in a neutral catholyte (pH ≈ 6.6) or in a strong acidic catholyte (Supplementary Figs. 6–9, pH = 1). The $H_2$ FE remained below 5% as $j$ increased up to 2.0 A/cm$^2$, whereas the CO FE remained as high as 90.0% at $j$ of 2.5 A/cm$^2$, resulting in a high $j_{CO}$ of 2.3 A/cm$^2$. Such $CO_2RR$ to CO performance in neutral catholyte distinguishes these electrocatalysts from other recent prominent electrocatalysts (Supplementary Fig. 6 and Supplementary Table 1). Subsequently, the CO FE decreased rapidly with increasing $j$, falling to 72.2% at 4.0 A/cm$^2$, indicating an increase in HER at further elevated $j$ values. Notably, During both $CO_2RR$ and HER, the consumption of $H^+$ will likely produce a locally alkaline environment close to the catalyst surface. As a result, we hypothesized that at high $j$ in a neutral electrolyte, rather than being reduced, a portion of the input $CO_2$ would instead be consumed in the electrolyte through a reaction with $OH^-$ to produce (bi)carbonate. Consequently, local $CO_2$ levels are inadequate and HER growth is kinetically more favorable.

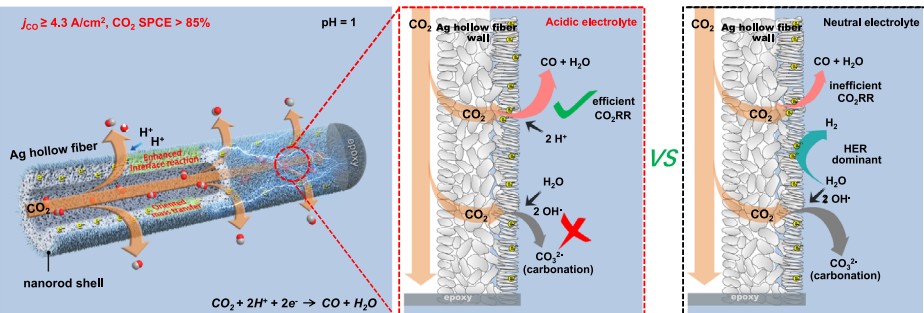

**Fig. 1 | Function outline of acidic $CO_2$ electroreduction over hollow fiber silver electrode.** Schematic of Ag hollow fiber penetration electrode for boosting $CO_2$ electroreduction to CO in a strongly acidic electrolyte (pH = 1) and the left schematic illustration of Ag hollow fiber was reproduced with permission from the reference[37]. Copyright [2022] [Springer Nature].

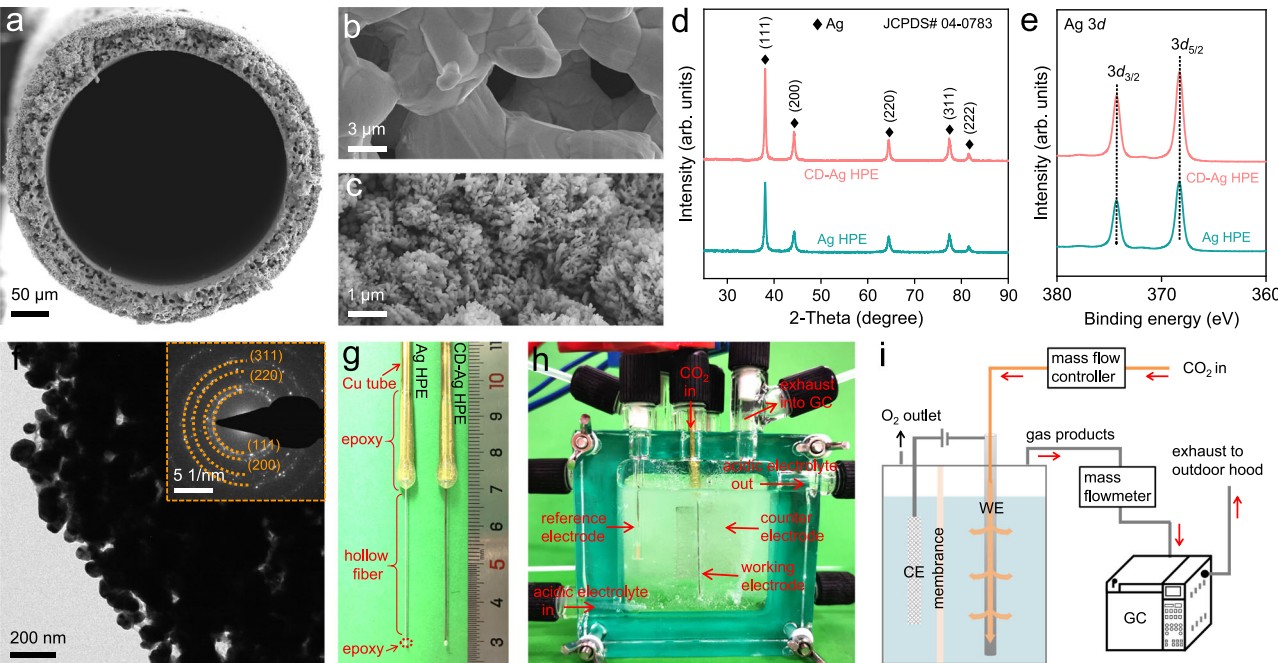

**Fig. 2 | Structural and compositional characterization.** SEM images of (**a**) cross section and (**b**), (**c**) outer surface of (**a**), (**c**) CD-Ag HPE and **b** Ag HPE. **d** XRD patterns, and **e** XPS spectra of Ag HPE and CD-Ag HPE. **f** TEM image and corresponding SAED pattern (insert of **f**) of CD-Ag HF. Optical images of the (**g**) working electrode of Ag HPE and CD-Ag HPE, (**h**) electrolyte flow two-compartment electrolysis cell from a side view. **i** Schematic illustration of the electrolysis system for CO₂ electroreduction. WE: working electrode, CE: counter electrode, GC: gas chromatograph.

## Effects of H⁺ concentration and CO₂ carbonation on CO₂RR

The calculations based on the reaction and diffusion of species model (Supplementary Fig. 10) within a typical diffusion layer of 50 μm showed that, in the presence of $K^+$ in a strong acidic electrolyte (pH = 1), the surface pH (distance to cathode of 0 μm) was similar to the bulk at $j < 100$ mA/cm² while became neutral or basic when current density increase further due to the consumption rate of local protons that exceeds mass transport of protons from the bulk (Fig. 3a, b, $j > 100$ mA/cm²), effective CO₂RR would dominate while HER was inhibited (Figs. 3c, d). Thus, the proton source of CO₂RR comes from water on the electrode surface, despite the bulk pH still being in an acidic range (Fig. 3a). As shown in Fig. 3a, although the pH at the cathode surface rapidly increased with the $j$ increase, the pH still remained in an acidic range with the increase in the distance from the cathode. When at a high $j$ of 1 A/cm², although the surface is alkaline, the pH decreased to 7 at 30 μm away from the cathode. In comparison, similar conditions (pH 7 and pH 10.5 at a distance to the cathode of 40 μm for bulk pH 4 and 7, respectively) were reached at much lower $j$ (about 200 mA/cm²) in electrolytes of pH 4 and 7 (Supplementary Fig. 11b, c). That means at high $j$, the carbonate formation would be more serious in an electrolyte with insufficient H⁺. In addition, the corresponding variation of CO₂ concentration distribution showed that, even in a strong acidic electrolyte with pH 1 (Fig. 3b), the surface available CO₂ concentration decreased gradually with the $j$ increased above 200 mA/cm², which is mainly due to the dual effects of the conversion of CO₂ and the carbonation of CO₂ caused by the increasing of pH[22,23]. However, the available CO₂ concentration increased to a high level within about 30 μm of the cathode, even at a high $j$ of 1 A/cm², mainly due to the rapid transport of sufficient H⁺ and CO₂ in the bulk electrolyte. In contrast, in the electrolytes with pH of 4 and 7 (Supplementary Figs. 12b, c), the available CO₂ concentrations began to decrease rapidly at a much lower $j$ and recover to a higher level at further distances from the cathode than that at pH 1, implying a greater CO₂ carbonation in an electrolyte with insufficient H⁺. Therefore, in order to pursue efficient CO₂RR and high CO₂ SPCE at high $j$, we

sought to conduct CO₂RR in strongly acidic electrolytes (pH 1), and attempted to explore, including the effect of CO₂ flow rate, H⁺ and K⁺ concentrations on it.

To overcome the problem of (bi)carbonate formation and minimize the effect of CO₂ carbonation under high $j$, we attempted to modulate the H⁺ concentration in the electrolyte (bulk pH) to neutralize the locally generated OH⁻, preventing the formation of (bi) carbonate. As shown in Fig. 3c, when $j \leq 0.5$ A/cm², the CO FE decreased with the pH decrease at the same $j$, whereas the H₂ FE exhibited the opposite trend (Fig. 3d and Supplementary Fig. 13). Additionally, the onset $j$ of CO₂RR to CO increased gradually with the decrease of pH from 3 to 0.5 (Fig. 3c). This can be attributed to the fact that with the decrease in pH, the local H⁺ concentration of the electrode increases rapidly, and the HER is more likely to dominate[40,41]. Thus, a higher value of $j$ is required to consume a substantial amount of H⁺ and modify the surface pH to make it more favorable for the kinetics of CO₂RR. That is the shift of onset $j$ for CO₂RR strongly depends on bulk electrolyte pH, which is consistent with the simulation results and other reports (Fig. 3a)[22-26]. When the applied $j$ was further increased from 0.5–2 A/cm², all the CO FEs exceeded 95% in the 4–1 pH range, and CO FE exceeded 80% at pH 0.5 (Fig. 3e and Supplementary Fig. 14). Interestingly, when the $j > 2$ A/cm², the CO FE increased as the pH decreases from 4 to 1 at the same $j$. This was irrespective of the decrease of all CO FE in the 4–0.5 pH range, which occurred as $j$ further increased. Notably, at pH 0, almost no CO₂RR was observed at any given $j$ (Figs. 3e, f and Supplementary Figs. 13f, 14f). This is because HER always dominated when the surface pH could not be modulated due to the local H⁺ consumption rate being significantly lower than the mass transport rate of the bulk H⁺ in the extremely acidic electrolytes[22,23].

Concurrently, the CO₂ carbonation of different pH electrolytes at different $j$ was further investigated (Fig. 4a and Supplementary Fig. 15). We discovered that in a near-neutral electrolyte (pH 4), the CO₂ carbonation percentage increased rapidly from 6.2% to 48.9% as $j$ increased from 0.5 to 4 A/cm². However, CO₂ carbonation decreased significantly as the pH of the electrolyte decreased. Accordingly, when

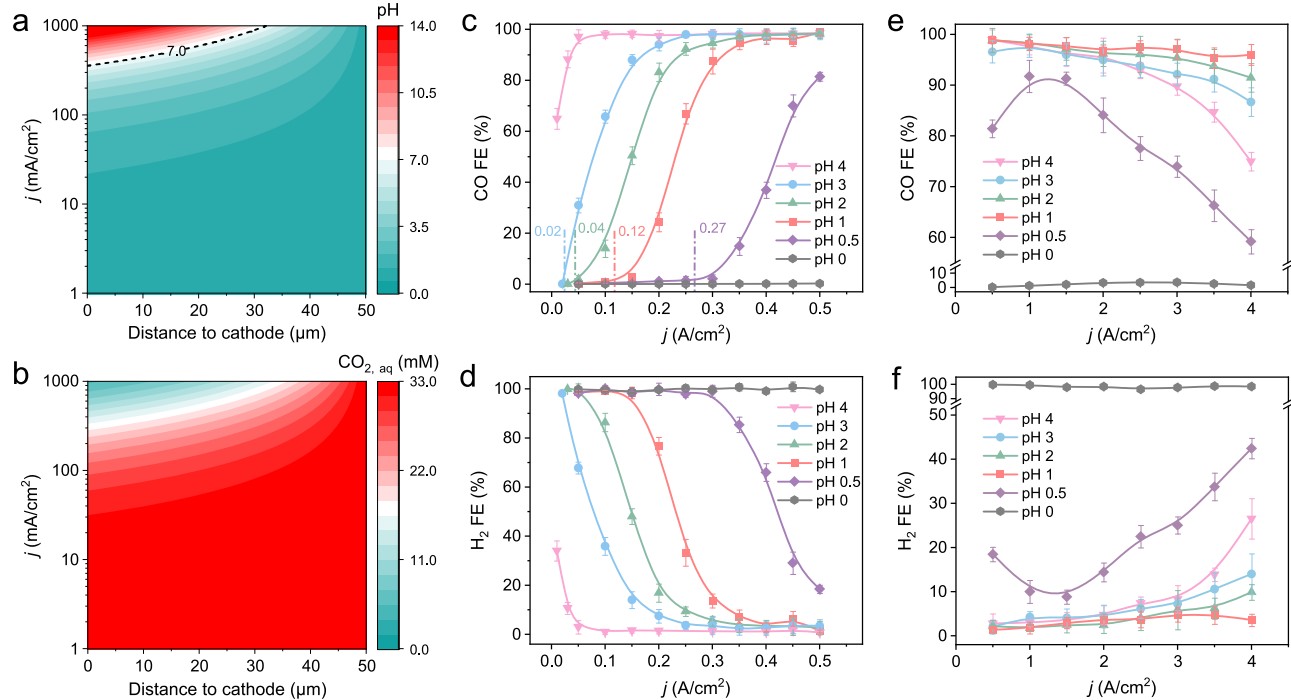

**Fig. 3 | Effects of H$^+$ concentration and CO$_2$ carbonation on CO$_2$RR.** Modeling of (**a**) pH, (**b**) concentration profile of CO$_2$ at different distances to cathode and $j$ in 0.05 M H$_2$SO$_4$ and 3 M KCl. **c, e** CO and **d, f** H$_2$ FE over CD-Ag HPE as a function of applied $j$ measured in 3 M KCl and H$_2$SO$_4$ catholytes with different pH values. The input CO$_2$ flow rate was 30 sccm. The error bars in (**c**)–(**f**) represent one standard deviation based on five independent tests.

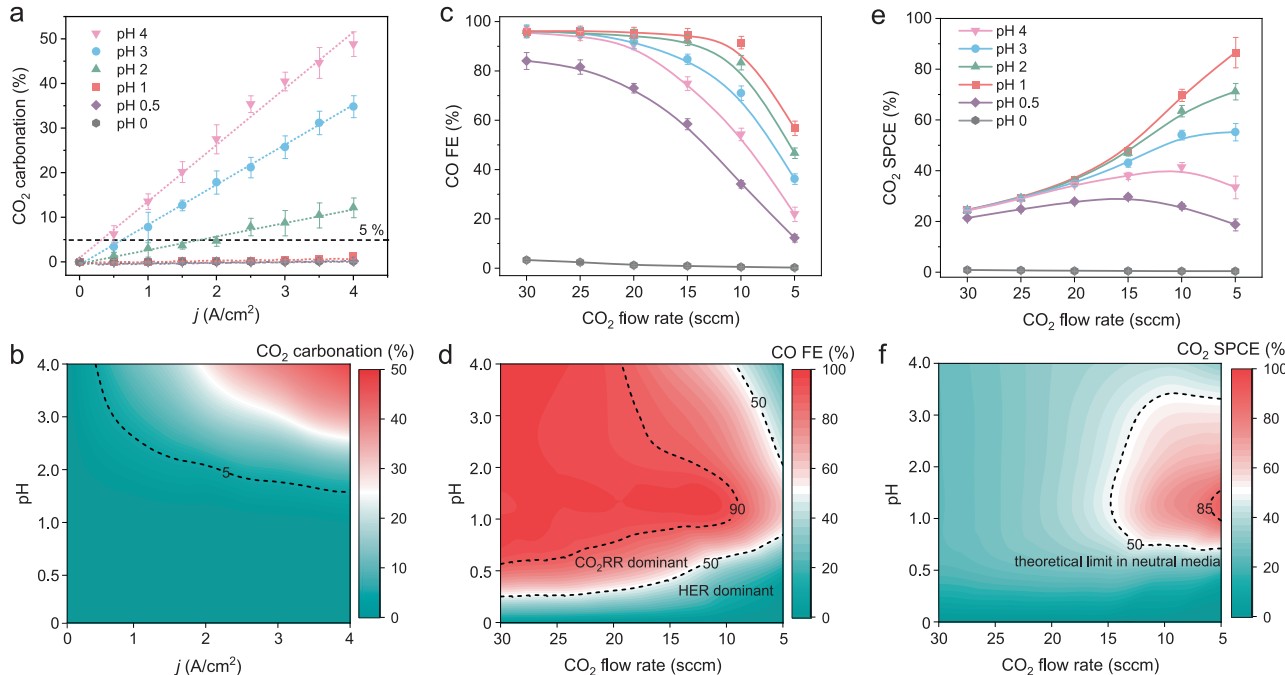

**Fig. 4 | Effects of H$^+$ concentration and CO$_2$ flow rate on CO$_2$RR. a** CO$_2$ carbonation percentage over CD-Ag HPE as a function of applied $j$ measured in 3 M KCl + H$_2$SO$_4$ catholytes with different pH values. **b** pH–$j$-dependent mapping distribution of CO$_2$ carbonation percentage over CD-Ag HPE. **c** CO FE and (**e**) CO$_2$ SPCE over CD-Ag HPE as a function of input CO$_2$ flow rate in 3 M KCl + H$_2$SO$_4$ catholytes with different pH values at a constant $j$ of 2 A/cm². The pH–CO$_2$ flow rate-dependent mapping distribution of (**d**) CO FE and (**f**) CO$_2$ SPCE over CD-Ag HPE at a constant $j$ of 2 A/cm². The error bars in (**a**), (**c**), (**e**) represent one standard deviation based on five independent tests.

pH ≤ 1, the CO$_2$ carbonation was always < 5%, even when $j$ reached 4 A/cm² (Fig. 4a). To visualize the change in the trend of CO$_2$ carbonation, the contour mapping distribution of CO$_2$ carbonation on a pH–$j$ plane (Fig. 4b) was plotted based on the date of Fig. 4a. It was clearly demonstrated that the CO$_2$ carbonation increased rapidly in tandem

with both the increase in $j$ and the bulk pH (Fig. 4b). Correspondingly, the region with the greatest CO$_2$ carbonation selectivity was located in the upper right of this map, indicating that the higher $j$ as well as higher pH, the more serious CO$_2$ carbonation. Based on the aforementioned findings, we reasoned that for an electrolyte with insufficient H$^+$

concentration, the local $OH^-$ was generated rapidly as $j$ increased. Some fraction of the input $CO_2$ would readily react with a large amount of locally generated $OH^-$ to form a (bi)carbonate coating on the surface of the electrode resulting in a significant reduction of local available $CO_2$ and $CO_2RR$ active sites[28,42]. As a result, when $CO_2RR$ at high $j$, local $CO_2$ and $CO_2RR$ active sites of the electrode were insufficient, and HER was kinetically more favorable. However, an acidic electrolyte with sufficient $H^+$ concentration can provide sufficient $H^+$ to maintain a moderate local pH, effectively prevent $CO_2$ carbonation, and ensure sufficient local $CO_2$ for high-efficiency $CO_2RR$ at high $j$ (Fig. 1).

### Effects of $H^+$ concentration and $CO_2$ flow rate on $CO_2RR$

To further confirm this hypothesis, we reduced the flow rate of the input $CO_2$ at a high $j$ of 2 A/cm² to make the effect of $CO_2$ carbonation on $CO_2RR$ more apparent. As depicted in Fig. 4c, when the input $CO_2$ flow rate was reduced from 30 to 25 sccm at pH 4, the $H_2$ FE remained > 3%, and the CO FE remained virtually unchanged (> 97%). This is because a high flow rate of input $CO_2$ could ensure relatively adequate local $CO_2$ for high-efficiency $CO_2RR$, even though ≈ 25% of $CO_2$ was lost (Fig. 4a). However, as the input $CO_2$ flow rate was further reduced from 25 to 5 sccm, the CO FE decreased rapidly (from 97% to 22%), while the $H_2$ FE increased dramatically (Fig. 4c and Supplementary Fig. 16a). Thus, it can be inferred that under the dual influence of $CO_2$ carbonation (Fig. 4a) and decreasing input $CO_2$ flow rate, the local $CO_2$ could not satisfy high-efficiency $CO_2RR$ at such a high $j$. In addition, the locally generated (bi)carbonate on the surface of the electrode may reduce the availability of $CO_2RR$ active sites[28,42]. Although the $CO_2$ SPCE gradually increased by decreasing the $CO_2$ flow rate to limit the availability of $CO_2$ (Fig. 4e and Supplementary Fig. 16i), it was still < 50% of the theoretical limit value for neutral electrolytes[22,28].

Then we reduced the input $CO_2$ flow rate of CD-Ag HPE in different pH electrolytes and observed the effect on HER (Supplementary Fig. 16h) and $CO_2RR$ (Fig. 4c). In the pH range 1–4 when the $CO_2$ flow rate was ≥ 20 sccm, the HER was effectively suppressed (Supplementary Fig. 16a–d), and the CO FE was always high (Fig. 4c; $FE_{CO}$ > 90%). However, when the pH was < 1, an excessively high local $H^+$ concentration increased $H_2$ FE significantly (Supplementary Fig. 16e, f)[22,23]. Moreover, as the $CO_2$ flow rate fell below 20 sccm, the CO FE decreased, and the $H_2$ FE increased dramatically. However, at identical low input $CO_2$ flow rates, CO FE had a volcano-like distribution in the 0–4 pH range. Consequently, at a high $j$ of 2 A/cm², the contour mapping distribution of CO FE on the pH–$CO_2$ flow rate plane clearly showed that, at a low $CO_2$ flow rate, the higher CO FE region was located at pH 1–2, in spite of CO FE is basically the same in pH 1–4 at high $CO_2$ flow rate (Fig. 4d). For the distribution of $CO_2$ SPCE (Fig. 4e), which was determined jointly by the $j_{CO}$ and input $CO_2$, the $CO_2$ SPCE increased rapidly at pH 1 and 2 with a decrease in the input $CO_2$ flow rate (from 20–5 sccm). Thus, when combined with the CO FE and input $CO_2$ flow rate, the contour mapping distribution of $CO_2$ SPCE on the pH–$CO_2$ flow rate plane directly showed that the most $CO_2$ SPCE selective area was located in the middle right of the map, where require both low input $CO_2$ flow rate and low pH of 1–2 (Fig. 4f). In other words, at a $CO_2$ flow rate of 5 sccm, the $CO_2$ SPCE at a pH of 1–2 exceeded 80%, which is roughly double that at pH 4 (Fig. 4e). In addition, the $j$–$CO_2$ flow rate-dependent mapping distribution of theoretical limit of CO FE based on $CO_2$ carbonation in different pH electrolyte also showed a higher theoretical limit of CO FE could be achieved only in a strong acidic electrolyte (Supplementary Fig. 17).

These observations indicated that a high flow rate of $CO_2$ can ensure high local $CO_2$ for high-efficiency $CO_2RR$ but limit $CO_2$ SPCE. However, in electrolytes with insufficient $H^+$ concentration, the $CO_2$ SPCE cannot be promoted effectively by merely reducing the input $CO_2$. It is worth noting that the locally generated (bi)carbonate on the surface of the electrode not only causes a reduction of locally available $CO_2$ but also covers the active sites for $CO_2RR$, resulting in the collapse

of CO FE at low input $CO_2$ flow rate. When $CO_2RR$ was conducted in an acidic electrolyte, the proper $H^+$ concentration in the electrolyte could effectively solve the problem of $CO_2$ carbonation and ensure that the local $CO_2$ concentration was sufficient despite a low input $CO_2$ flow rate. Therefore, while $CO_2$ availability is constrained, high-efficiency $CO_2RR$ can be maintained by reducing the input $CO_2$, effectively promoting the $CO_2$ SPCE.

### $K^+$ effect on acidic $CO_2RR$

Given the essential role of alkali cation in the activation of $CO_2$ and inhibition of HER[22,24,25], we first performed linear voltammetry curve (LSV) analysis (Fig. 5a). In 0.05 M $H_2SO_4$ (pH 1) without any $K^+$ electrolyte, the voltammetric properties of the CD-Ag HPE hardly changed regardless of the surrounding atmosphere (Ar or $CO_2$), indicating that only HER occurs. However, HER activity was significantly suppressed by $K^+$ presentation, concurrently, $CO_2RR$ occurred. This may be attributed to the fact that the presence of $K^+$ in the acidic electrolyte shielded the electric field in the diffusion layer of the cathode and reduced the $H^+$ concentration around the active site, which not only suppressed the HER, but also stimulated $CO_2$ activation and conversion. (Supplementary Fig. 18a)[22,24,41]. Subsequently, we conducted Tafel analysis in 0.05 M $H_2SO_4$ + KCl catholytes at pH 1 but with different $K^+$ concentrations (Fig. 5b). On the one hand, the CO Tafel slope was found to decrease with increasing $K^+$ concentrations, reaching a minimum (104 mV dec⁻¹) at 3 M $K^+$. This result suggests that the rate-determining step (RDS) for CO formation comprised the adsorption of $CO_2$, which could be altered with the $K^+$ concentrations[22,43,44]. In other words, the activation energy barrier of the electron transfer over the electrode may be reduced in an electrolyte with a high $K^+$ concentration, which is consistent with the faster initial one-electron transfer step required to form an adsorbed *$COO^-$ intermediate. On the other hand, it can be seen that with the increase of $K^+$ concentration, the changing trend of $H_2$ Tafel slope value was opposite to that of CO. That is, the $H_2$ Tafel slope value increased with increasing of $K^+$ concentrations (Supplementary Figs. 19b, c). This is consistent with the results of LSV (Fig. 5a), indicating that the presence of $K^+$ would suppress the HER. In addition, the total Tafel slope values of CO and $H_2$ gradually decreased with increasing of $K^+$ concentrations (Supplementary Fig. 19c), which was consistent with the results in Fig. 5c, where a faster electron transfer was verified by the lowest interfacial charge transfer resistance ($R_{ct}$) of CD-Ag HPE in 3 M $K^+$ (0.9 Ω cm², Fig. 5c and Supplementary Table 8). In addition, the higher effective electric double layer capacitance ($C_{dl}$) of CD-Ag HPE in acidic electrolyte with high $K^+$ concentration, which was correlated with the electric field strength an enhanced electric field trend (Fig. 5c and Supplementary Table 8). These results are consistent with the hypothesis, that is the hydrated $K^+$ physisorbed on the cathode in the acidic electrolyte modify the distribution of electric field in the double layer, which not only impedes HER by suppression of migration of $H^+$, but also promotes $CO_2$ reduction by stabilization of key intermediates (Supplementary Fig. 18)[22,24,41].

We also examined the $CO_2RR$ product distribution in 0.05 M $H_2SO_4$ (pH 1) with varying $K^+$ concentrations at different $j$ (Fig. 5d, e and Supplementary Fig. 20). Even at a high $j$ of 4 A/cm², almost no CO could be detected in 0.05 M $H_2SO_4$ without $K^+$ (Fig. 5d and Supplementary Fig. 20a), which was consistent with the LSV results (Fig. 5a). In addition, the HER selectivity (Fig. 5e) decreased as the $K^+$ concentration increased, while the $CO_2RR$ selectivity (Fig. 5d) increased for all given values of $j$. Particularly, at a constant high $j$ of 2 A/cm², the CO FE increased steadily from 61.2% with 0.1 M $K^+$ to 96.8% with 3 M $K^+$. The contour mapping distribution of CO FE on the $j$–$K^+$ concentration plane further directly showed that, $CO_2RR$-dominated regions require the presence of high concentrations of $K^+$, while obtaining high CO FE at high $j$ is more dependent on high $K^+$ concentrations (Fig. 5f). In addition, the porous micro/nanostructured Ag (Supplementary Fig. 21)

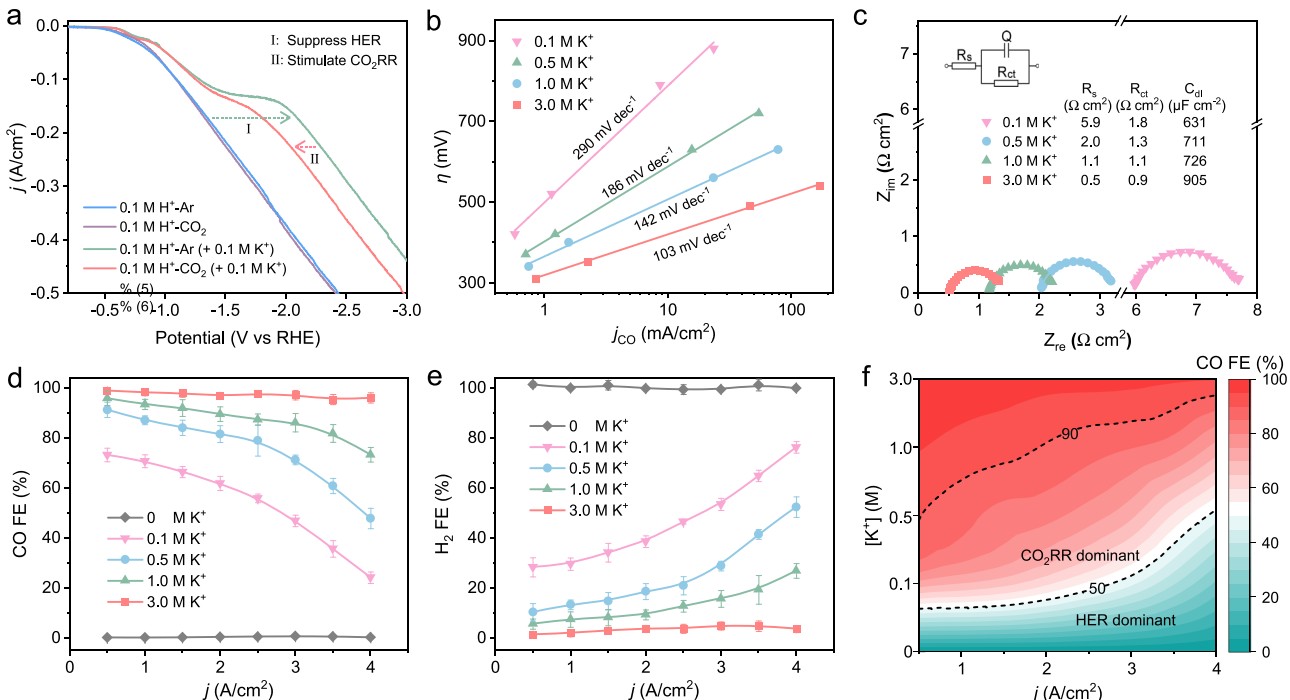

**Fig. 5 | K$^+$ effect on acidic CO$_2$RR. a** LSV curves of CD-Ag HPE in Ar-saturated or CO$_2$-saturated pure H$_2$SO$_4$ or H$_2$SO$_4$ with 0.1 M K$^+$ catholytes at pH 1. **b** Tafel slopes and **c** EIS Nyquist plots obtained in catholytes with different K$^+$ concentrations at pH 1. **d** CO and (**e**) H$_2$ FE over CD-Ag HPE as a function of applied $j$ measured in H$_2$SO$_4$ catholytes with different K$^+$ concentrations at pH 1 (input CO$_2$ flow rate: 30 sccm). **f** $j$ − K$^+$ concentration-dependent mapping distribution of CO FE over CD-Ag HPE (input CO$_2$ flow rate: 30 sccm). The error bars in (**d**), (**e**) represent one standard deviation based on five independent tests.

was found to be conducive to increasing the local K$^+$ concentration[22,45,46], which can be attributed to the amplified electric field near the pore sites, thereby inhibiting HER and promoting the activity of CO$_2$RR (Supplementary Fig. 22). Note that both controlled experiments and DFT calculations showed that, in an acidic electrolyte, the effect of anions on CO$_2$RR reactivity was not significant; substitution of Br$^-$, SO$_4^{2-}$ or PO$_4^{3-}$ for Cl$^-$ showed product distribution similar to that of the Cl$^-$ case (Supplementary Fig. 23), and the Gibbs free energy of *COOH and *H, the key intermediates from CO$_2$RR to CO and HER, respectively, basically did not change whether there was Cl$^-$ adsorption or not (Supplementary Figs. 24, 25).

**Effects of CO$_2$ flow rate and $j$ on acidic CO$_2$RR**

To pursue a high CO$_2$ SPCE under high CO$_2$RR to CO activity, we analyzed the input CO$_2$ flow rate effect on acidic CO$_2$RR product distribution at various $j$ (Fig. 6a and Supplementary Fig. 26). In strong acidic electrolytes, there is almost no loss of CO$_2$ and CO$_2$RR active sites because, in the absence of (bi)carbonate precipitation, the activation of high K$^+$ concentration and the high input flow rate of CO$_2$ can guarantee sufficient CO$_2$ and CO$_2$RR active sites locally on the catalyst surface for high-efficiency CO$_2$RR. Consequently, the CO FE could maintain a high value (FE$_{CO}$ > 95%) even when the $j$ reached 4 A/cm$^2$ (Fig. 6a and Supplementary Fig. 26). As the input flow rate decreased, the local CO$_2$ supply became relatively insufficient, and as $j$ increased, the CO FE began to decline rapidly. Thus, combined with the CO FE and $j$, the most $j_{CO}$ selective region of the $j$−CO$_2$ flow rate-dependent mapping distribution of CO FE is located in the top left corner, where the high flow rate of input CO$_2$ could maintain the sufficient CO$_2$ supply to achieve high CO FE at high $j$ due to the unique structure of the HPE (Fig. 6b). These results indicate that a high input CO$_2$ flow rate is necessary for CD-Ag HPE to achieve high-efficiency CO$_2$RR to CO under high $j$ in acidic electrolyte. The detailed performances of the CD-Ag HPE at high CO$_2$ flow rate (30 sccm) revealed that the H$_2$ FE remained < 5% as the $j$ increased, up to 4.5 A/cm$^2$, while the CO FE remained as high as 95.06%, yielding 4.28 A/cm$^2$ $j_{CO}$ at −1.41 V vs. RHE

(Fig. 6c and Supplementary Fig. 27). The corresponding CO yield and CO energy efficiency were 80.83 mmol/(h cm$^2$) and 48.25%, respectively (Supplementary Fig. 27 and Supplementary Table 2). Such performance stands out among recently reported prominent electrocatalysts for CO formation from CO$_2$RR (Supplementary Fig. 28 and Supplementary Table 1).

Regarding CO$_2$ SPCE, it is determined by both the $j_{CO}$ and input CO$_2$ flow rate. At a high input CO$_2$ flow rate of > 20 sccm, the CO FE of CD-Ag HPE could remain above 80% even as $j$ increased to 4 A/cm$^2$ (Fig. 6a), whereas the CO$_2$ SPCE increased only very slowly as $j$ was increased (Fig. 6d, e). However, a moderate CO$_2$ flow rate and a substantial CO FE at high $j$ were more conducive to achieving a high CO$_2$ SPCE. This trend was further visualized in the the $j$-CO$_2$ flow rate-dependent mapping distribution of CO$_2$ SPCE (Fig. 6e). The most CO$_2$ SPCE selective region was located in the middle right of the map, where high CO$_2$ SPCE of > 80% was only achieved when the $j$ is around 2 A/cm$^2$ and the flow rate is less than 10 sccm. Particularly, when the CO$_2$ flow rate was decreased from 30−5 sccm at a constant high $j$ of 2 A/cm$^2$, the CO$_2$ SPCE increased from 22% to 87% for CO$_2$RR to CO, thereby exceeding the theoretical limit of 50% in neutral/alkaline systems. This was comparable to the record level of 90% at low $j$ of < 0.2 A/cm$^2$ in acidic systems (Fig. 6f and Supplementary Table 6)[28].

Although highly challenging, the long-term operation of electrocatalysts under high current density is crucial for their practical applications[3,14]. Correspondingly, the $j$ was fixed at 2 A/cm$^2$, and the flow rate was set to 10 sccm in the acidic electrolyte stability test. During the 200-h continuous test, the CO FE remained consistently above 90%, while the corresponding CO$_2$ SPCE fluctuated around 70% (Fig. 6g). In addition, the surface structures remained highly stable after the stability test (Supplementary Fig. 29), and the postreaction XRD and XPS further revealed the stable compositions of CD-Ag HPE after electrolysis (Supplementary Fig. 30 and Supplementary Table 10), which were responsible for the steady CO$_2$ electroreduction performance. Thus, the overall CO$_2$RR performance is well-placed

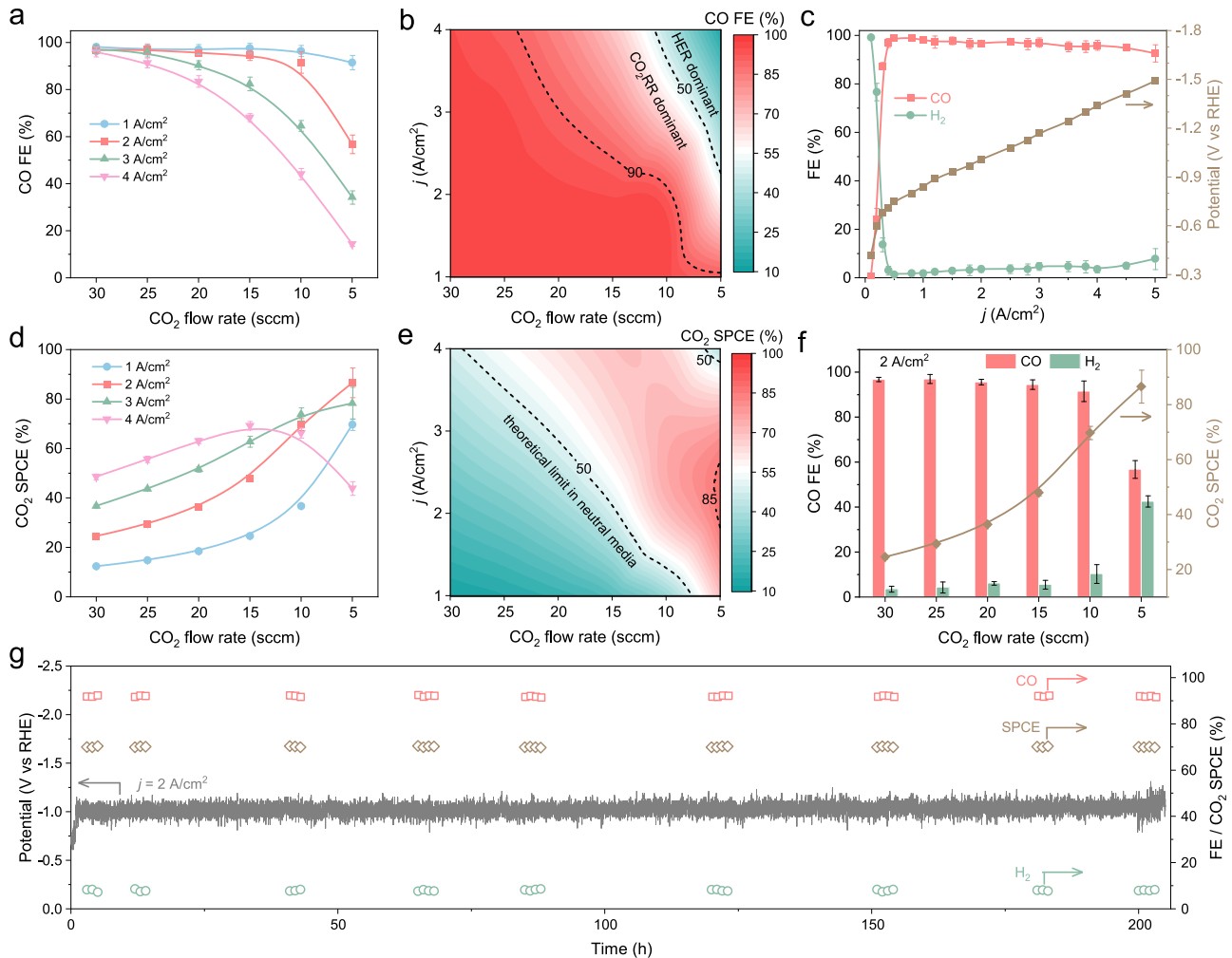

**Fig. 6 | Effects of CO₂ flow rate and *j* on acidic CO₂RR. a** CO FE and (**d**) CO₂ SPCE over CD-Ag HPE as a function of input CO₂ flow rate measured in 3 M KCl + 0.05 M H₂SO₄ catholytes (pH 1) at different applied *j*. The *j*–CO₂ flow rate-dependent mapping distribution of (**b**) CO FE and (**e**) CO₂ SPCE over CD-Ag HPE in 3 M KCl + 0.05 M H₂SO₄ catholytes (pH 1). **c** CO, H₂ FE and potential over CD-Ag HPE as a function of applied *j* measured in 3 M KCl + 0.05 M H₂SO₄ catholytes (pH 1, input CO₂ flow rate: 30 sccm). **f** CO, H₂ FE and CO₂ SPCE over CD-Ag HPE as a function of input CO₂ flow rate measured in 3 M KCl + 0.05 M H₂SO₄ catholytes (pH 1) at a constant *j* of 2 A/cm². **g** Long-term performance at a constant *j* of 2 A/cm² in 3 M KCl + 0.05 M H₂SO₄ catholytes (pH 1, input CO₂ flow rate: 10 sccm). The error bars in (**a**), (**c**), (**d**), (**f**) represent one standard deviation based on five independent tests.

among recently reported outstanding electrocatalysts for CO formation from CO₂ reduction, including *j*, CO FE, CO yield (95% CO FE and 80.83 mmol/(h cm²) CO yield at high *j* of 4.5 A/cm²), CO₂ SPCE and stability (87% CO₂ SPCE and 200 h long-term test at *j* of 2 A/cm²) (Supplementary Figs. 26–31 and Supplementary Tables 1, 2, 6), demonstrating great potential for scalable application. And at the current density as high as 4.5 A/cm², the overpotential of CD-Ag HPE was only 1.3 V, which was comparable to the high-current density electrocatalyst (Supplementary Table 1). In order to demonstrate the scalability for practical applications using hollow fiber penetration electrodes, single-, 2-, 5- and 10-tube arrays of CD-Ag HPE were further adopted and tested in a 2-electrode acidic system (pH = 1). All CD-Ag HPE array electrodes with different tube numbers showed highly similar FE distributions of CO and H₂ at given high *j* range (Supplementary Figs. 32, 33). Thus, the *j*_CO over these CD-Ag HPE array electrodes also exhibited almost same rapidly growth trend with increasing *j*. Although CO FE and *j*_CO over the CD-Ag HPE array electrodes slightly decreased with increasing tube number and *j*, 10-tube CD-Ag HPE array still possessed over 90% of CO FE at high *j* of 4 A cm⁻². These results implied the potential scalability for practical applications using hollow fiber penetration electrodes.

## Theoretical calculations

The promoting mechanism of K⁺ and H⁺ in the aqueous microenvironment surrounding the Ag active site of the CD-Ag HPE was further simulated using DFT calculations. The formation of adsorbed COOH (*COOH) and H (*H) intermediates at the active site were thought to be the RDS for CO₂RR and HER (Supplementary Figs. 34, 35)[46–48], respectively. Thus, we first simulated an Ag (111) surface with a K⁺ concentration ranging from 0 to 3 (1/18 per H₂O molecule), and compared the Gibbs free energy (G) change difference for the formation of *COOH and *H (Fig. 7a). Although both the G(*COOH) and G(*H) decreased with increasing K⁺ concentration, the G(*COOH−*H) also decreased rapidly (from 1.21 eV to 0.60 eV) with the increase of K⁺. This was consistent with the experimental findings, and the results indicated that an increase in K⁺ concentration could promote both CO₂RR and HER, with CO₂RR being the more dominant effect. On this basis, we further calculated the G for the formation of *COOH and *H at different *H coverages ranging from 0 to 3 (1/9 per site) (Fig. 7b). Correspondingly, we found volcano-shaped relationships between G(*COOH) and *H coverage, which is in stark contrast to the result that G(*H) increases as *H coverage increases. Consequently, the G(*COOH−*H) reached a minimum of 0.04 eV in the optimized

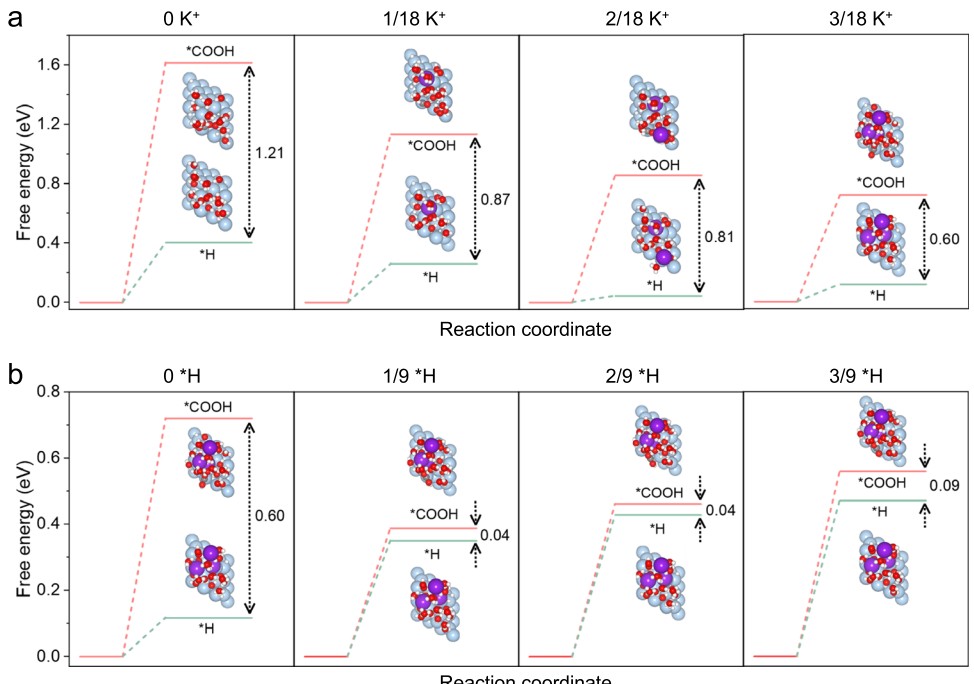

**Fig. 7 | Theoretical calculations.** The free energy for the formation of *COOH (G(*COOH)), *H (G(*H)), and their energy difference (G(*COOH)-G(*H)) on Ag (111) plane at various (**a**) $K^+$ concentrations (1/18 per $H_2O$ molecule) and (**b**) *H coverages (1/9 per site). The light blue, purple, red, gray and white balls represent Ag, K, O, C and H, respectively.

structures with a high $K^+$ concentration and moderate *H coverage (3/18 $K^+$−1/9 *H and 3/18 $K^+$−2/9 *H). In addition, we explored the effect of different hydrogen sources on a model of 3 $K^+$ (1/18 / $H_2O$ molecule) (Supplementary Figs. 35–38). It was found that the values of G (*COOH), G (*H) and G (*COOH−*H) calculated by the $H_3O^+$ model were basically the same as those calculated by the *H model, and they were all much lower than the model values of $H_2O$ as hydrogen source (Supplementary Fig. 39). Taken together, these DFT results suggest that the coexistence of $H^+$ and $K^+$ could promote the $CO_2RR$ synergistically. A high $K^+$ concentration and a moderate *H coverage are likely to facilitate *COOH formation for $CO_2RR$ over *H formation for HER, which is consistent with the experimentally observed high $CO_2RR$ to CO conversion in strongly acidic media with a high $K^+$ concentration.

Combined with controlled experiments and theoretical studies, we demonstrated that the presence of $K^+$ in the acidic electrolyte controlled the onset of the $CO_2RR$, and that a moderate concentration of $H^+$ effectively prevented the carbonation of $CO_2$ and $CO_2RR$ active sites due to the precipitation of (bi)carbonate, ensuring sufficient $CO_2$ and $CO_2RR$ active sites at the catalyst surface for high-efficiency $CO_2RR$ at ampere-level current density. Thus, by optimizing the $K^+$ and $H^+$ concentration and $CO_2$ flow rate in a strong acidic electrolyte, a high CO FE > 95% at 4.5 A/cm² and >200 h of stability testing at 2 A/cm² are achieved. In addition, by limiting the availability of input $CO_2$, the $CO_2$ SPCE for $CO_2RR$ reached 87% at a high $j$ of 2 A/cm², demonstrating remarkable $CO_2$ conversion capability. This study, therefore, provides a means for high-efficiency $CO_2$ conversion in strong acid by modulating catalyst microenvironments, with great potential for practical application.

## Methods

### Chemicals and materials

Ag powder (99.9%, 50 nm) was purchased from Ningbo Jinlei Nano Materials Co., Ltd. Polyetherimide (PEI) was purchased from Saudi Basic Industries Corporation (SABIC). N-Methyl-2-pyrrolidone (NMP), sulfuric acid ($H_2SO_4$), potassium sulfate ($K_2SO_4$), potassium

bicarbonate ($KHCO_3$), potassium chloride (KCl) and potassium hydroxide (KOH) were purchased from Sinopharm Chemical Reagent Co., Ltd. Nafion 117 proton exchange membranes (PEMs) were purchased from DuPont. All chemicals were used as received without further purification. Electrolyte solutions were prepared using 18.2 MΩ $H_2O$ (ultrapure water, from Master-S30UVF water purification system).

### Catalyst preparation

Ag hollow fiber (Ag HF) was fabricated by a combined phase-inversion/sintering process (Supplementary Fig. 1)[37]. Briefly, commercially available polyetherimide (PEI, 24 g) was added to N-Methyl-2-pyrrolidone (NMP, 96 g), followed by ultrasonic treatment for 1 h to obtain a homogeneous and transparent solution. Then Ag powder (80 g) was added to the above solution. The as-obtained mixture was further treated by the planetary ball-milling (using 250 mL zirconia jar and φ5 mm zirconia balls) at 300 rpm for 24 h to form a uniform slurry. After cooling to room temperature, the slurry was vacuumed (1 mbar) for 5 h to remove bubbles and then to obtain a casting solution. Next, the casting solution was extruded through a spinning machine and shaped in a water bath via the phase-inversion process. After spinning, the as-formed tubes were kept in a water bath for 24 h to eliminate the solvent completely, followed by stretching and drying in ambient conditions with a humidity of ~28% for 48 h to obtain a green body. The green body was cut into appropriate lengths and then calcined in an airflow (100 mL/min) at 600 °C (heating rate: 1 °C/min) for 6 h to remove PEI. After being naturally cooled to room temperature, the calcined green body was then reduced in a 5% $H_2$ (argon balance) flow (100 mL/min) at 300 °C (heating rate: 1 °C/min) for 3 h to obtain Ag HF.

The Ag HF with an exposed length of 4 cm was stuck into a copper tube using conductive silver adhesive for electrical contact (see Fig. 2g for details), while the end of the Ag HF tube as well as the joint between the Ag HF and copper tube were sealed and covered with gas-tight and nonconductive epoxy. After drying at room temperature for 12 h, a working Ag HF penetration electrode (Ag HPE) was obtained with an exposed geometric area of 0.5 cm² (S = $\pi D_{out}L$ = 3.14 × 400 ×

$10^{-4} \times 4 \approx 0.5$ cm$^2$, where S is the electrode area, D$_{out}$ is the outer diameter of hollow fiber, and L is the length of hollow fiber).

Ag$_2$CO$_3$-Ag HPE was synthesized from Ag HPE by electrochemical redox activation treatments. Typically, the Ag HPE was subjected to oxidation and reduction treatments on a Biologic VMP3 potentiostat using a three-electrode system in a gas-tight two-compartment electrolysis cell containing a Nafion 117 membrane as the separator, a KCl-saturated Ag/AgCl reference electrode and a platinum mesh (3 cm × 3 cm) counter electrode. The electrolyte solution was CO$_2$-saturated 0.5 M KHCO$_3$, and the CO$_2$ flow rate was kept at 2 mL/min. Prior to the experiments, the electrolysis cell was vacuumized and then purged with CO$_2$ for 30 min. The Ag HPE was electrochemically oxidized at a fixed potential of 2.0 V (vs. Ag/AgCl) for 4 min to obtain Ag$_2$CO$_3$-Ag HPE. Subsequently, the Ag$_2$CO$_3$-Ag HPE was reduced at a fixed potential of −0.50 V (vs. Ag/AgCl) for 10 min to obtain CD-Ag HPE. The CD-Ag HPE possessed the same exposed geometric area of 0.5 cm$^2$ (S=πD$_{out}$L = 3.14 × 400 × 10$^{-4}$ × 4 = 0.5 cm$^2$). For the 10-tube CD-Ag HPE array electrode, the exposure geometric area was 5 cm$^2$ (S = nπD$_{out}$L = 10 × 3.14 × 400 × 10$^{-4}$ × 4 = 5 cm$^2$, where n is the number of hollow fiber tubes). The electrochemical oxidation reaction and reduction reaction obeyed Eqs. (1) and (2), respectively.

$$2Ag + 2H_2O + HCO_3^- \rightarrow Ag_2CO_3 + 6e^- + O_2 \uparrow + 5H^+ \qquad (1)$$

$$Ag_2CO_3 + 5H^+ + 6e^- \rightarrow 2Ag + HCO_3^- + 2H_2 \uparrow \qquad (2)$$

In addition, the OD-Ag HPE was also treated in the Ar-saturated 0.5 M KOH with the same electrochemical redox activation treatments as that of CD-Ag HPE. That is, the Ag HPE that underwent 60 s of oxidation and 600 s of reduction in Ar-saturated 0.5 M KOH electrolyte solution, respectively.

## Material characterization

The cross-section and surface morphologies of samples were observed via scanning electron microscopy (SEM) with a SUPRRATM 55 microscope using an accelerating voltage of 5.0 kV. Transmission electron microscopy (TEM) investigations were conducted with a JEM-ARM300F microscope operated at 300 kV. X-ray diffraction (XRD) measurements were performed on a Rigaku Ultima 4 X-ray diffractometer using a Cu Kα radiation source (λ = 1.54056 Å) at 40 kV and 40 mA. X-ray photoelectron spectroscopy (XPS) tests were conducted using a Quantum 2000 Scanning ESCA Microprobe instrument with a monochromatic Al Kα source (1486.6 eV). The binding energies in all XPS spectra were calibrated according to the C 1 s peak (284.8 eV).

## Electrochemical measurements

All electrochemical measurements were performed using a Biologic VMP3 potentiostat in a two-compartment electrolysis cell at an ambient temperature and pressure. The electrolysis cell comprises two symmetrical compartments made of quartz glass with an inner height of 5.0 cm, an inner length of 5.0 cm and an inner width of 1.5 cm (Fig. 2g, h). The cathodic and anodic compartments were separated by a Nafion 117 membrane, and the electrolysis cell was equipped with a KCl-saturated Ag/AgCl reference electrode in the cathodic compartment and a platinum mesh (3 cm × 3 cm). CO$_2$-saturated H$_2$SO$_4$ containing various concentrations of KCl or different pH was used as catholyte, and 0.5 M K$_2$SO$_4$ + 0.05 M H$_2$SO$_4$ aqueous solution was used as anolyte. Prior to the experiments, the electrolysis cell was vacuumed, and then CO$_2$ was continuously delivered into the cathodic compartment at a constant rate of 30 mL/min for 30 min. During all of the electrochemical measurements, the CO$_2$ flow rate was fixed at 30 mL/min. Note that the input CO$_2$ flow rate was fixed at 30 mL/min during all electrochemical measurements, but the exhaust flow rate was not 30 mL/min at all due to the hydrogen evolution and CO$_2$ consumption. Thus, the outlet flow rate of the electrolysis cell was measured by an independent Alicat® mass flowmeter (Fig. 2i and Supplementary Table 4). The cathodic electrolyte was 3.0 KCl + 0.05 M H$_2$SO$_4$, the anodic electrolyte was 0.5 M K$_2$SO$_4$ + 0.05 M H$_2$SO$_4$, unless otherwise stated.

For electrochemical characterizations, the electrochemical impedance spectroscopy (EIS) measurements were performed at −0.2 A/cm$^2$ with a voltage amplitude of 50 mV, and the frequency limits were typically set in the range from 50 Hz to 500 kHz. The effective double-layer capacitance (Cdl) was obtained from the constant phase element (CPE) parameters and the two resistances using the Brug formula[37,49]:

$$C_{dl} = T^{\frac{1}{p}} \left( \frac{1}{R_s} + \frac{1}{R_{ct}} \right)^{\frac{P-1}{P}} \qquad (3)$$

Where R$_s$ is the solution resistance, R$_{ct}$ is the charge transfer resistance, T is CPE constant and P is CPE exponent. For the CO$_2$ electroreduction tests, a Biologic VMP3 potentiostat was used for small current situations (≤ 400 mA), while the Biologic VMP3 potentiostat was connected to a VMP3 booster chassis with a 10 A current option to be used under large current situations (≥ 400 mA). The CO$_2$RR performance of the CD-Ag HPE array electrodes with different tube numbers in the 2-electrode acidic system (pH = 1) was performed using an ANS6050D (50 V, 50 A) direct-current source from ANS Power Co., Ltd. with constant current control applied. In addition, both the anolyte and catholyte were cycled in anodic and cathodic compartments with a fixed flow rate of 50 mL/min by using two identical peristaltic pumps (JIHPUMP BT-50EA 153YX). For the long-term performance test of CO$_2$ electroreduction, the current density was fixed at −2 A/cm$^2$ in CO$_2$-saturated 3 M KCl + 0.05 M H$_2$SO$_4$ catholyte, the exhaust from the cathodic compartment was measured by online GC during the entire 200-hour test.

All the current in the main text and supplementary materials were geometrically normalized to the electrode area. All the applied potentials were recorded against the KCl-saturated Ag/AgCl reference electrode and then converted to those versus the reversible hydrogen electrode (RHE) with iR corrections using the following equation:

$$E(\text{vs. RHE}) = E(\text{vs. Ag/AgCl}) + 0.197V + 0.0591V \times \text{pH} + 0.85iR_s \qquad (4)$$

Where E (vs. Ag/AgCl) is the applied potential, pH indicates the H$^+$ concentrations of the electrolyte solutions (Supplementary Table 7), i is the current density at each applied potential, and R$_s$ is the solution resistance obtained via EIS measurements (Supplementary Table 8). In order to avoid the overcorrected potentials, 85% iR correction was applied as the previous reports[17,48]. All applied potentials in the main text and Supplementary Information are referred to as RHE, unless otherwise stated. Note that the XY data at different pH values of Figs. 4a, c, e were first converted into XYZ data by the origin® software, to obtain the corresponding contour maps of Figs. 4b, d, f, respectively. Moreover, the XY data at different K$^+$ concentrations of Fig. 5d in manuscript were first converted into XYZ data in origin® software, to obtain the corresponding contour map of Fig. 5f. In addition, the XY data at different current densities of Figs. 6a, d were first converted into XYZ data by the origin® software, to obtain the corresponding contour maps of Fig. 6b, e, respectively. Taking Fig. 4b for example, the X represents current density of Fig. 4a, Y represents pH of Fig. 4a, and Z represents percentage loss of CO$_2$ of Fig. 4a, which were then standard smoothed and transformed back into a virtual matrix. The as-obtained virtual matrix was further presented in the form of a contour map, i.e., Fig. 4b, in consistency with the previous reports[28,50].

## Product quantifications

Gas-phase products from the cathodic compartment were directly vented into a gas chromatograph (GC-2014, Shimadzu) equipped with a Shincarbon ST80/100 column and a Porapak-Q80/100 column using a flame ionization detector (FID) and a thermal conductivity detector (TCD) during the electroreduction tests and analyzed online. FID was used for CO quantification (as well as $CH_4$, $C_2H_4$ and $C_2H_6$), while TCD was used for $H_2$ and CO quantification. In addition, the concentration of unreacted $CO_2$ in the outlet gas was analyzed with an independent Agilent 7890B gas chromatograph (Supplementary Fig. 9). A thermal conductivity detector (TCD) with a carbon molecular sieves column (TDX-1) was used for $CO_2$ quantification. All faradaic efficiencies reported were based on at least three different GC runs, and the error bars of all figures in this work were based on the standard deviations of at least five independent electrochemical tests, unless otherwise stated. High-purity argon (99.999%) was used as the GC carrier gas. In all the $CO_2$ electrolysis tests, only $H_2$ and CO were the gas-phase products, and their faradaic efficiencies were calculated as follows:

$$FE = \frac{C_{product} \times 10^{-6} \times v_{outlet} \times 10^{-3} \times t \times n \times F}{V_m \times Q} \times 100\% \quad (5)$$

where $C_{product}$ is the concentration of the gas-phase products (ppm), $v_{outlet}$ is the outlet flow rate of the electrolysis cell. Note that the exhaust flow rate of the electrolysis cell was not equal to the input $CO_2$ flow rate at all due to the hydrogen evolution and $CO_2$ consumption. Thus, the outlet flow rate of electrolysis cell was measured by an independent Alicat® mass flowmeter (Fig. 2i). The actual measured outlet flow rate was higher than the inlet flow rate due to HER occurrence (Supplementary Table 3 and Fig. 5f). Thus the actual exhaust outlet flow rates (corrected $v_{outlet}$) were measured by the Alicat® mass flowmeter (Fig. 2i), which possessed a full scale of 50 sccm with the accuracy ± (0.8% of Reading + 0.2% of Full Scale), as shown in the note d of Supplementary Table 3. That is the experiment errors caused by the Alicat® mass flowmeter could be negligible under the $CO_2$ electroreduction conditions. In addition, the Alicat® mass flowmeter results were calibrated using a mixture (including CO, $H_2$, $CO_2$ and water vapor compositions at room temperature) that approximates the actual outlet gas composition (Supplementary Table 4). t is the reaction time, n is the number of transferred electrons for producing CO or $H_2$, F is the Faraday constant, $V_m$ is the gas mole volume, and Q is the total quantity of the electric charge. The CO formation rate was calculated using the following equation:

$$CO\ formation\ rate = \frac{Q \times FE_{CO}}{F \times n \times t \times S} \quad (6)$$

Where S is the geometric area of the electrode ($cm^2$).

The $j_{CO,limit(gas)}$ is the theoretical limit of CO partial current density with all gas-phase $CO_2$ molecules input into the electrolysis cell were reduced to CO products. The theoretical limits of CO product partial current density, i.e., $j_{CO,lim(gas)}$ were calculated using the following equation[37,38]:

$$j_{CO,lim(gas)} = \frac{nF}{S} \frac{v_{CO_2}}{V_m} \quad (7)$$

So, the theoretical limits of CO FE, i.e., $FE_{CO,lim(gas)}$ were calculated by the following equation:

$$FE_{CO,lim(gas)} = \frac{j_{CO,lim(gas)}}{j_{total}} \times 100\% \quad (8)$$

The $CO_2$ SPCE, the $CO_2$ carbonation and unreacted $CO_2$ were calculated as follows:

$$SPCE = \frac{produced\ CO}{Input\ CO_2} \times 100\% \quad (9)$$

$$CO_2\ carbonation = \frac{Input\ CO_2 - Unreacted\ CO_2 - Converted\ CO_2}{Input\ CO_2} \times 100\% \quad (10)$$

$$Unreacted\ CO_2 = V_{outlet} - Proceed\ CO - Proceed\ H_2 \quad (11)$$

$$Converted\ CO_2 = Proceed\ CO \quad (12)$$

Where the input $CO_2$ ($V_{input}$) of electrolysis cell was controlled by a Alicat® mass flow controller, and the outlet flow rate ($V_{outlet}$) of electrolysis cell was measured by an independent Alicat® mass flowmeter. The actual outlet gas of electrolysis cell contains unreacted $CO_2$, produced CO (from $CO_2$RR) and produced $H_2$ (from the HER). The produced CO and $H_2$ at given current density could be quantified by online GC. Furthermore, to verify our calculation of unreacted $CO_2$, the concentration of unreacted $CO_2$ in the outlet gas was measured by an independent online GC (Fig. 2i).

By assuming that the overpotential of oxygen evolution reaction on the anode side is zero, the cathodic energy efficiency for CO was calculated as follows[12]:

$$EE_{CO} = \frac{(1.23 + (-E_{CO})) \times FE_{CO}}{1.23 + (-E)} \quad (13)$$

Where $E_{CO}$ is −0.11 V (vs. RHE); 1.23 V is the thermodynamic potential for water oxidation in the anode side.

Possible liquid-phase products from the cathodic compartment after $CO_2$ electrolysis for 1 h were analyzed using another off-line GC-2014 (Shimadzu) equipped with a headspace injector and an OVI-G43 capillary column (Supelco, USA). No liquid-phase product was detected by the off-line GC. The post-reaction catholyte solution was also analyzed by a 600 MHz NMR spectrometer (Bruker) for possible liquid-phase products (especially formate and acetate). After an hour of electrolysis, an aliquot of catholyte solution (0.5 mL) was mixed with 0.1 mL $(CH_3)_3Si(CH_2)_3SO_3Na$ (DSS) (6 mM) and 0.1 mL $D_2O$ for use as internal standards. No liquid-phase product was detected by $^1H$ NMR (Supplementary Figs. 6–9).

## COMSOL multiphysics simulations

A reaction-diffusion model (Supplementary Fig. 10) was used to simulate the local pH and $CO_2$ concentration using COMSOL Multiphysics software in a typical 50 μm diffusion layer[22,23,25]. All the interactions between species in the electrolyte ($CO_2$, $HCO_3^-$, $CO_3^{2-}$, $SO_4^{2-}$, $K^+$, $Cl^-$, $OH^-$, $H^+$ and $H_2O$) were considered (Supplementary Table 11). Specifically, one end of the one-dimensional simulation area is set as the working electrode surface, and the other side is set as the bulk concentration to describe the bulk electrolyte. We used Henry's law to calculate the $CO_2$ concentration, assuming that the $CO_2$ fugacity is 1 bar.

$$C_{CO2,aq}^0 = K_H^0 C_{CO2,gas}^0 \quad (14)$$

$K_H^0$ is the Henry's constant, which can be calculated by using the equation below, where T is the temperature.

$$\ln K_H^0 = 93.457 \times \frac{100}{T} - 60.2409 + 23.3585 \times \ln \frac{T}{100} \quad (15)$$

Due to the high concentration of the ions, the saturated concentration of $CO_2$ in an electrolyte is corrected using the following equations.

$$\log\frac{C^0_{CO2,aq}}{C_{CO2,aq}} = K_s C_s \tag{16}$$

$$K_s = \sum (h_{ion} + h_G) \tag{17}$$

$$h_G = h_{G,0} + h_T(T - 298.15) \tag{18}$$

$C_s$ is the molar concentration and $K_s$ is the Sechenov's constant.

We considered the following homogeneous and heterogenous reactions in our model, which are based on the previously published works[22,23,25]. The heterogenous reactions take place in the electrolyte as follow:

$$2H_2O + 2e^- \rightarrow H_2 + 2OH^- \tag{19}$$

$$CO_2 + H_2O + 2e^- \rightarrow CO + 2OH^- \tag{20}$$

In this work, two sets of Butler-Volmer boundary conditions ($CO_2$RR and HER) were set to represent the two main sets of reactions on electrode surface, the kinetic parameters sets for these two reactions were the Tafel slopes of CO (103 mV dec$^{-1}$) and $H_2$ (56 mV dec$^{-1}$), respectively, and the FEs of CO and $H_2$ in the model were calculated and simulated based on these parameters.

Over the whole domain, the following homogenous reactions occur:

$$CO_2 + H_2O \rightleftharpoons H^+ + HCO_3^- \tag{21}$$

$$HCO_3^- \rightleftharpoons H^+ + CO_3^{2-} \tag{22}$$

$$CO_2 + OH^- \rightleftharpoons HCO_3^- \tag{23}$$

$$HCO_3^- + OH^- \rightleftharpoons CO_3^{2-} + H_2O \tag{24}$$

$$H_2O \rightleftharpoons H^+ + OH^- \tag{25}$$

The bulk concentrations and pH values were measured experimentally and implemented in the model. The thickness of the diffusion layer was assumed to be 50 μm. The electrode surface undergoes a reduction reaction, its current characteristics follow the Bulter-Volmer equation and the Nernst equation, its potential characteristics follow the Nernst equation, the concentration of the opposite body phase is set to the corresponding initial concentration, and the ion migration in the simulation area follows the Nernst Planck equation. The solution process is based on the MUMPS (multiple massively parallel spark direct solver) steady-state solver, and the relative tolerance and residual factor are set to 1E$^{-8}$ and 1, respectively, eight layers of boundary layer subdivision are set on the simulated electrode surface to ensure the accuracy of the simulation results. The pH and $CO_2$ concentration distribution near the electrode surface was calculated by solving the operating current from 0–1000 mA/cm$^2$ at bulk pH of 1, 4, and 7, respectively.

The electrode surface reaction follows the BV equation:

$$i_{loc} = i_0 \left( \exp\left(\frac{\alpha_a F n \eta}{RT}\right) - \exp\left(\frac{-\alpha_c F n \eta}{RT}\right) \right) \tag{26}$$

Where $i_{loc}$ is the local current density at the electrode/electrolyte interface, $i_0$ is the exchange current density, $\alpha_c$ and $\alpha_a$ is the cathodic and anodic charge transfer coefficients, $\eta$ is the activation overpotential.

The balance potential follows the Nernst equation:

$$E_{eq} = -\frac{\triangle G}{nF} \tag{27}$$

$$E_{eq} = E_{eq,ref} - \frac{RT}{nF} \ln \prod_i \left(\frac{a_i}{a_{i,ref}}\right)^{\nu_i} \tag{28}$$

Where $E_{eq}$ is the electrode potential, $\Delta G$ is the Gibbs free energy, $E_{eq,ref}$ is the standard electrode potential, $a_i$ is the (electrode reactive ion concentration), $a_{i,ref}$ is the (standard electrode reactive ion concentration), $\nu_i$ is the reaction stoichiometric number.

The transfer of dilute substances follows Fick's law:

$$N_i = J_i = -D_i \nabla c_i \tag{29}$$

$$\frac{\partial c_i}{\partial t} + \nabla N_i = R_{i,tot} \tag{30}$$

Where $J_i$ is the ion flux, $D_i$ is the diffusion coefficient ($D_{Li} = 1 \times 10^{-7}$ m$^2$s$^{-1}$), $c_i$ is the concentration of ion, $\nabla c_i$ is concentration gradient.

The transportation of electricity follows the Nernst Planck relationship:

$$N_i = -D_i \nabla c_i - z_i u_{m,i} F c_i \nabla \varnothing_l = J_i + u c_i \tag{31}$$

Where $z_i$ is the transfer number ($z_{Li} = 1$), $u_{m,i}$ is the electric mobility coefficient, $\phi$ is the electrolyte potential.

## Computational theoretical calculations

The present first principle DFT calculations are performed by Vienna Ab initio Simulation Package with the projector augmented wave method[51]. The exchange function is treated using the generalized gradient approximation of Perdew-Burke-Emzerh functional[52]. For Kohn–Sham wave functions, the cutoff energy of the corresponding plane-wave basis set was set to 450 eV. The K points meshing was obtained from the Monkhorst-Pack scheme[53]. Grimme's DFT-D3 methodology was used to describe the dispersion interactions. A vacuum width of of 15 Å along the Z axis was created to ensure negligible interaction. The force convergence criterion was set to 0.02 eV/Å and energy convergence criterion was 10$^{-4}$ eV. To fully consider the solvation effect, 18 explicit water molecules were optimized, and a local minimum via the hydrogen bond network was formed. The Gibbs free energy change (ΔG) of each step is calculated using the following formula:

$$\triangle G = \triangle E + \triangle ZPE - T\triangle S \tag{32}$$

where ΔE is the electronic energy difference directly obtained from DFT calculations, ΔZPE is the zero point energy difference, T is the room temperature (298.15 K) and ΔS is the entropy change.

## Data availability

The data supporting the findings of the study are available within the paper and its Supplementary Information. Source data are provided in this paper.

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

## Acknowledgements

This work was financially supported by the Ministry of Science and Technology of China (National Key R&D Program of China, 2022YFA1504604), the Strategic Priority Research Program (A) of the Chinese Academy of Sciences (XDA0390400), the National Natural Science Foundation of China (nos. 91745114, 21802160, 22302223), the Hundred Talents Program of Chinese Academy of Sciences (no. 2060299), the Youth Innovation Promotion Association of the Chinese Academy of Sciences (no. E224301401), Shanghai Excellent Principal Investigator (no. 23XD1404400), Shanghai Sailing Program (nos. 23YF1453300, 18YF1425700), Science and Technology Innovation Pla-nof Shanghai Science and Technology Commission (Nos. 23DZ1202600.23DZ1201804), the Outstanding Young Talent Project of Shanghai Advanced Research Institute, the Chinese Academy of Sciences (no. E254991ZZ1), the Foundation of Key Laboratory of Low-Carbon Conversion Science & Engineering, Shanghai Advanced Research Institute, Chinese Academy of Sciences (no. KLLCCSE-202207Z, SARI, CAS), Shanghai Functional Platform for Innovation Low Carbon Technology, and the Major Project of the Science and Technology department of Inner Mongolia (no. 2021ZD0020).

## Author contributions

S.L., X.D., G.W., and W.C. conceived the research and designed the experiment. S.L., X.D., G.W., Y.S., J.M., A.C., C.Z., G.L., Y.W., X.L., and J.W. conducted the experiment. S.L., X.D., G.W. analyzed the data. S.L., X.D., G.W., Y.S., G.L., and W.C. discussed and interpreted the results. W.C. and W.W. supervised the project. S.L., X.D., and G.W wrote the manuscript with input from all authors. All authors revised the manuscript.

## Competing interests

The authors declare no competing interests.
