## [Peer Review File · Nature Communications]

REVIEWER COMMENTS

Reviewer #1 (Remarks to the Author):

I believe that this paper will eventually become a good paper in Nature. The results are exciting and certainly worth publishing, but the paper needs considerable revision.

- 1) The present paper was written as a specialist paper with all of the key figures in the appendix. It assumed that I knew what their system was like, and focused on highlighting the results in the appendix. The paper needs to instead be focused on a general audience. What are you measuring, what are the key findings,
- 2) When I read the paper, I could not tell what their experimental system was like. I am guessing that the authors are feeding CO₂ into a hollow fiber placed in an acidic media but it was not clear from the paper.
- 3) It was not clear how the authors are collecting the CO.
- 4) The paper included several interesting figures, but these are largely ignored in the text. These need to be referred to and explained.
- 5) The bulk of the present draft is a discussion of the results in the supplemental material. This text should be eliminated. As a reader I could not understand most of the text without laying the main body and supplemental information side by side. The manuscript needs to be accessible without having to continuously refer to the supplementary material.

Reviewer #2 (Remarks to the Author):

Reviewer #3 (Remarks to the Author):

In this manuscript, Chen et al. demonstrate the use of a hollow fiber silver penetration electrode to achieve a high CO FE in an acidic solution. In the article, the authors conduct experiments, coupled with continuum-level mass transfer simulations and density functional theory calculations. The article is well-written and pedagogical; I enjoyed reading through it. To be acceptable for Nature Communications, however, there are some major issues that should be addressed. Note that this reviewer is coming from a modeling and simulation background, so the points that I'll focus on will mainly be related to that.

- On page 6, a 1D Nernst-Planck model is presented to describe the pH and CO₂ concentration as a function of distance from the cathode. While this type of model is certainly common for solid planar electrodes, it is not clear how this is related to the configuration presented. Based on Fig. 1, my understanding is that gaseous CO₂ is supplied and flows through the porous electrode, where reaction occurs near the many triple-phase boundaries. How is this model consistent with the electrolyzer configuration? If it is not, how can the model be modified to address this?

- How were the parameters used in the Butler Volmer equations derived?

- In the discussion of the K⁺ cation, there is no mention of how the electrical double layer is affected; the structure of this would certainly change the microenvironment near the catalyst as K⁺ concentration changes. Can the authors discuss this, and comment on why this is relevant or not?

- Related to the above point, EIS measurements are presented as a function of K⁺ concentration. Can information about the double layer be extracted based on this?

- Through the DFT simulations, would it be possible to construct a microkinetic model of the kinetics using the measured G's?

- In the DFT simulations, how do the concentrations of K⁺ relate to concentrations of K⁺ in the electrolyte (e.g., in units of mM)?

Reviewer #4 (Remarks to the Author):

Shoujie Li et al. describe in their manuscript entitled "Ampere-level CO₂ electroreduction with single-pass conversion exceeding 85% in strong acid over a silver penetration electrode" very promising experimental results for the electrochemical CO₂ reduction employing a newly developed silver hollow fiber penetration electrode. Traditionally, alkaline electrolytes are used during the electrochemical reduction of CO₂ to CO over silver catalysts. These electrolytes lead to a strong reduction of the carbon efficiency due to the simultaneously occurring homogeneous reactions of CO₂ with hydroxide ions, typically resulting in a CO₂ utilization of approx. 50%. Therefore, it has been

suggested to employ acidic electrolytes, which would enhance the carbon efficiency. On the other hand, the hydrogen evolution reaction is strongly favored at low pH values. For these reasons proper electrode designs and management of the local pH conditions at the silver catalyst are required. The presented results show very promising Faradaic efficiencies and single pass conversions, even at extremely high current density. Acidification of the KCl electrolyte with sulfuric acid towards a pH of around 1 led to the best performance and it could be clearly shown that the presence of potassium ions is required to achieve a high Faradaic efficiency. The experimental findings are also supported by simulations of the local electrolyte composition close to the electrode through a reaction-diffusion model as well as by DFT calculations. Overall, this is a very interesting and valuable contribution which can be published in Nature Communications after careful consideration of the following comments:

- In the literature overview, I miss a recent publication by Löffelholz et al. (DOI: 10.1149/1945-7111/ad0eba) with quite similar trends of the FE towards CO, although over a very different electrode and at lower current density.

- I recommend moving the details about the geometrical area of the hollow fiber electrode from the SI to the main manuscript, as it is very important for interpretation of the results. This would facilitate the understanding of the reader.

- The authors used KCl acidified with H₂SO₄ as electrolyte. However, the DFT calculations show no significant effect of Cl⁻ when compared to SO₄²⁻. This raises the question why KCl was used instead of K₂SO₄, introducing an anion species to the system that would possibly lead to undesirable anode reactions in a technical systems with mixed anolyte and catholyte.

- The authors claim that the CO₂ loss by homogeneous reactions is negligibly small and justify this with experimental and simulation results. However, the experimental procedure for determination of the CO₂ loss should be described in greater detail. If I see it correctly, the gas-chromatographic method did not include the determination of carbon dioxide. Was the calculation based solely on the flow rate measurements? How precisely could the carbon balance be closed? I also recommend to use the more common term “carbon efficiency” instead of “CO₂ loss” to represent results for CO₂ conversion during electrochemical vs. homogeneous reactions.

- The results of the stability test over 200 hours are very promising. However, silver is not stable in acidic environment and Löffelholz et al. (DOI: 10.1149/1945-7111/ad0eba) found a relatively rapid deterioration of their electrode performance in contact with acidified potassium sulfate electrolyte. Can the authors explain this difference?

- The authors should describe in more detail how the iR correction was performed. At the current densities and gas flows through the fiber, intense bubbling can heavily impact the inner resistance of the electrolyte gap between WE and RE. How was this taken into account? Moreover, why was an 85% iR correction chosen instead of the more common 90% or full correction?

- In the experimental setup, a flat Pt-mesh is used as counter electrode. Could this result in an inhomogeneous distribution of potential in the working electrode between the side positioned towards the CE and the side facing away from it?

- Can the authors posit about the applicability of this type of electrode in large-scale systems?

- In the manuscript and supplementary material, formatting is often inconsistent (e.g., jco,lim(gas) vs. jco,lim(gas)). These inconsistencies should be fixed to improve the overall appeal of the publication. Additionally I recommend rephrasing of some unusual formulations like “pH is the Pondus Hydrogenii value”.

- Page 6, line 6 from bottom: typing error “50 mm”.

Responses to Reviewer 1

Comment: I believe that this paper will eventually become a good paper in Nature. The results are exciting and certainly worth publishing, but the paper needs considerable revision.

Response: We thank the reviewer very much for the positive appraisal and valuable suggestions to improve our manuscript. The manuscript and Supplementary Information have been revised according to these valuable suggestions.

Comment 1: The present paper was written as a specialist paper with all of the key figures in the appendix. It assumed that I knew what their system was like, and focused on highlighting the results in the appendix. The paper needs to instead be focused on a general audience. What are you measuring, what are the key findings,

Response: We highly appreciate the reviewer's valuable suggestions to improve our manuscript. The manuscript and Supplementary Information have been thoroughly revised according to these valuable suggestions. In order to show the electrode and experimental system in detail, some key structural and compositional characterizations of hollow fiber penetration electrode and experimental system in previous Supplementary Information have been moved to revised manuscript and made into Fig. 2. Furthermore, several key figures including Figs. 4d, 4f, 5f, 6b and 6e (corresponding to Figs. 3d, 3f, 4f, 5b and 5e in previous manuscript) have been fully explained and discussed in the revised manuscript. In addition, the detailed discussions about Supplementary Figure 17 (corresponding to Supplementary Figure 16 in previous manuscript) have been moved to the revised Supplementary Information, and the relevant results have also been largely condensed in the revised manuscript. After these movements, the manuscript looks clearer and readily understood.

Comment 2: When I read the paper, I could not tell what their experimental system was like. I am guessing that the authors are feeding CO₂ into a hollow fiber placed in an acidic media but it was not clear from the paper.

Response: We thank the reviewer very much for the thoughtful suggestions. We are sorry that we did not present our electrode and experimental system clearly in the previous manuscript. According to these valuable suggestions, some key characterizations of hollow fiber penetration electrode and experimental system in previous Supplementary Information have been moved to the revised manuscript as Fig. 2. The related descriptions about electrode and experimental system have been also added in revised manuscript as follows:

“The electrochemical experiment of the CO₂RR was conducted in two chamber electrolysis cell with a three-electrode system at room temperature, where the CD-Ag HPE was used as the working electrode and gas diffuser (Fig. 2g-i). During CO₂ electroreduction, CO₂ penetrated through the porous wall of the CD-Ag HPE into the electrolytes via the copper tube, forming a large amount of bubbles. This unique

oriented mass transfer of CO₂ could induce the in-situ formation of extensive dynamic CO₂(gas)–liquid–catalyst triphasic reaction interfaces, which significantly improve the mass transfer of CO₂, electrons, protons, products as well as CO₂RR kinetics³⁴⁻³⁹. Subsequently, a mixture of CO produced by CO₂RR, H₂ produced by HER, and unreacted CO₂ flows out through an outlet connected to the top right of the electrolysis cell. The actual outlet flow rate was measured by an independent mass flowmeter and then sent to online gas chromatography (GC) for quantification. And the exhaust from the GC was vented to the outdoor hood (Fig. 2h, i).”.

Fig. 2 | Structural and compositional characterization. **a**, SEM images of outer surfaces of **a**, Ag HPE and **c**, CD-Ag HPE, **b**, cross section of CD-Ag HPE. **d**, XRD patterns, and **e**, XPS spectra of Ag HPE and CD-Ag HPE. **f**, TEM image and corresponding SAED pattern (insert of **f**) of CD-Ag HF. Optical images of the **g**, working electrode of Ag HPE and CD-Ag HPE, **h**, electrolyte flow two-compartment electrolysis cell from a side view. **i**, Schematic illustration of the electrolysis system for CO₂ electroreduction.

Comment 3: *It was not clear how the authors are collecting the CO.*

Response: We thank the reviewer very much for the careful reading and insightful suggestion to improve our manuscript. We are sorry that we did not clarify how CO and H₂ were collected and measured in our experimental system in the previous version. The schematic illustrations about the gas products (CO, H₂) measurements as well as the exhaust from the gas chromatography (GC) have been supplemented in Fig. 2i. The relevant descriptions have been added into the revised manuscript as follows:

“The electrochemical experiment of the CO₂RR was conducted in two chamber electrolysis cell with a three-electrode system at room temperature, where the CD-Ag HPE was used as the working electrode and gas diffuser (Fig. 2g-i). During CO₂ electroreduction, CO₂ penetrated through the porous wall of the CD-Ag HPE into the electrolytes via the copper tube, forming a large amount of bubbles. This unique

oriented mass transfer of CO₂ could induce the in-situ formation of extensive dynamic CO₂(gas)–liquid–catalyst triphasic reaction interfaces, which significantly improve the mass transfer of CO₂, electrons, protons, products as well as CO₂RR kinetics³⁴⁻³⁹. Subsequently, a mixture of CO produced by CO₂RR, H₂ produced by HER, and unreacted CO₂ flows out through an outlet connected to the top right of the electrolysis cell. The actual outlet flow rate was measured by an independent mass flowmeter and then sent to online gas chromatography (GC) for quantification. And the exhaust from the GC was vented to the outdoor hood (Fig. 2h, i).”.

Comment 4: The paper included several interesting figures, but these are largely ignored in the text. These need to be referred to and explained.

Response: We thank the reviewer very much for the careful reading and constructive suggestions to improve our manuscript. We are sorry that several key figures (e.g. Figs. 3b, 3d, 3f, 4f, 5b and 5e in previous manuscript) were not discussed in detail in the previous manuscript. According to these constructive suggestions, several key figures including Figs. 4d, 4f, 5f, 6b and 6e (corresponding to Figs. 3d, 3f, 4f, 5b and 5e in previous manuscript) have been fully explained and discussed in the revised manuscript as follows:

“... To visualize the change in the trend of CO₂ carbonation, the contour mapping distribution of CO₂ carbonation on a pH-*j* plane (Fig. 4b) was plotted based on the data of Fig. 4a. It was clearly demonstrated that the CO₂ carbonation increased rapidly in tandem with both the increase in *j* and the bulk pH (Fig. 4b). Correspondingly, the region with the greatest CO₂ carbonation selectivity was located in the upper right of this map, indicating that the higher *j* as well as higher pH, the more serious CO₂ carbonation.”.

“...Consequently, at a high *j* of 2 A/cm², the contour mapping distribution of CO FE on the pH–CO₂ flow rate plane clearly showed that, at low CO₂ flow rate, the higher CO FE region was located at pH 1–2, in spite of CO FE is basically the same in pH 1-4 at high CO₂ flow rate (Fig. 4d).”.

“... Thus, when combined with the CO FE and input CO₂ flow rate, the contour mapping distribution of CO₂ SPCE on the pH–CO₂ flow rate plane directly showed that the most CO₂ SPCE selective area was located in the middle right of the map, where require both low input CO₂ flow rate and low pH of 1-2 (Fig. 4f).”.

“... The contour mapping distribution of CO FE on the *j*–K⁺ concentration plane further directly showed that, CO₂RR-dominated regions require the presence of high concentrations of K⁺, while obtaining high CO FE at high *j* is more dependent on high K⁺ concentrations (Fig. 5f).”.

“... Thus, combined with the CO FE and *j*, the most *j*_{CO} selective region of the *j*–CO₂ flow rate-dependent mapping distribution of CO FE is located in the top left corner, where the high flow rate of input CO₂ could maintain the sufficient CO₂ supply to achieve high CO FE at high *j* due to the unique structure of the HPE (Fig. 6b).”.

“... However, a moderate CO₂ flow rate and a substantial CO FE at high *j* were more conducive to achieving a high CO₂ SPCE. This trend was further visualized in the the *j*-CO₂ flow rate-dependent mapping distribution of CO₂ SPCE (Fig. 6e). The most

CO₂ SPCE selective region was located in the middle right of the map, where high CO₂ SPCE of >80% was only achieved when the j is around 2 A/cm² and the flow rate is less than 10 sccm.”.

Comment 5: The bulk of the present draft is a discussion of the results in the supplemental material. This text should be eliminated. As a reader I could not understand most of the text without laying the main body and supplemental information side by side. The manuscript needs to be accessible without having to continuously refer to the supplementary material.

Response: We highly appreciate the reviewer for the careful reading and valuable suggestions to improve our manuscript. According to these valuable suggestions, in order to show the electrode and experimental system in detail, some key characterization of hollow fiber penetration electrode and experimental system have been integrated into Fig. 2 in the revised manuscript. In addition, the detailed discussions about Supplementary Figure 17 (corresponding to Supplementary Figure 16 in previous manuscript) have been moved to the revised Supplementary Information, and the relevant results have also been largely condensed in the revised manuscript as follows:

“... In addition, the j -CO₂ flow rate-dependent mapping distribution of theoretical limit of CO FE based on CO₂ carbonation in different pH electrolyte also showed a higher theoretical limit of CO FE could be achieved only in a strong acidic electrolyte (Supplementary Fig. 17).”.

“To further investigate the impact of the concentrations of H⁺ and local availability of CO₂ on CO₂RR performance at high j of CD-Ag HPE, the j -CO₂ flow rate-dependent mapping distribution of theoretical CO FE limit based on CO₂ carbonation in different pH electrolyte were plotted. As shown in the Supplementary Figure 17, at a high flow rate of 30 mL/min, the theoretical CO FE limit values of all pH electrolytes (0-4) could basically reach or approach 100% even at j as high as 4 A/cm², which is consistent with the experimental results (Fig. 3e, f). However, compared with the theoretical CO FE limit values of pH 1, 0.5 and 0 with basically no CO₂ carbonation (Supplementary Figure 17d-f), the theoretical CO FE limit values of pH 4, 3 (Supplementary Figure 17a, b) rapidly decreases with the decreasing of input CO₂ flow rates and the increasing of j . In particular, at pH 4, when the $j > 2$ A/cm², as long as the CO₂ flow rates <25 sccm, the theoretical CO FE limit values were <90% (Supplementary Figure 17a). The theoretical CO FE limit values were even less than 50% at the j higher than 3 A/cm², where the HER dominated. In contrast, at a pH of 1 (Supplementary Figure 17d), higher theoretical CO FE limit values could be achieved at lower input CO₂ flow rates and higher j , indicating higher theoretical limit j_{CO} as well as a higher CO₂ SPCE could be achieved only in a strong acidic electrolyte.”.

Responses to Reviewer 2

***Comment:** I co-reviewed this manuscript with one of the reviewers who provided the listed reports. This is part of the Nature Communications initiative to facilitate training in peer review and to provide appropriate recognition for Early Career Researchers who co-review manuscripts.*

Response: We thank the reviewer very much for participating in the peer-review process as a co-reviewer. We also thank the reviewer very much for the valuable suggestions to improve our manuscript. The manuscript and Supplementary Information have been revised according to these valuable suggestions.

Responses to Reviewer 3

Comment: *In this manuscript, Chen et al. demonstrate the use of a hollow fiber silver penetration electrode to achieve a high CO₂ FE in an acidic solution. In the article, the authors conduct experiments, coupled with continuum-level mass transfer simulations and density functional theory calculations. The article is well-written and pedagogical; I enjoyed reading through it. To be acceptable for Nature Communications, however, there are some major issues that should be addressed. Note that this reviewer is coming from a modeling and simulation background, so the points that I'll focus on will mainly be related to that.*

Response: We appreciate the reviewer's positive opinions. We also thank the reviewer very much for the careful reading and valuable suggestions to improve our manuscript. The manuscript and Supplementary Information have been revised according to these valuable suggestions.

Comment 1: *On page 6, a 1D Nernst-Planck model is presented to describe the pH and CO₂ concentration as a function of distance from the cathode. While this type of model is certainly common for solid planar electrodes, it is not clear how this is related to the configuration presented. Based on Fig. 1, my understanding is that gaseous CO₂ is supplied and flows through the porous electrode, where reaction occurs near the many triple-phase boundaries. How is this model consistent with the electrolyzer configuration? If it is not, how can the model be modified to address this?*

Response: We highly appreciate the reviewer for the valuable comment to improve our manuscript. The reviewer is right that gaseous CO₂ is supplied and flows through the porous hollow fiber electrode, where reaction occurs near the many triple-phase boundaries. Actually, from a macroscopic point of view, the outer surface of the hollow fiber electrode is tubular, thus, the triple-phase reaction interface is approximately equal to the annular outer surface of the hollow fiber, which is indeed different from that of the solid planar electrode. However, from a microscopic point of view, the triple-phase reaction interface of hollow fiber electrode could be differentiated into numerous planar reaction micro-regions, which are not intrinsically different from the reaction micro-regions of the solid planar electrode (Supplementary Figure 10). Therefore, the 1D Nernst-Planck model could be applied to the current electrode configuration to describe the local pH and CO₂ concentration as a function of distance from the cathode, regardless of the configuration of the electrode itself. We will explore some optimized models that take into account the configuration of the electrode in the near future to more accurately simulate the real reaction interface.

Supplementary Figure 10. Schematic of a one-dimensional (1D) COMSOL models of **a**, working electrode of hollow fiber penetration electrode and **b**, solid planar electrode.

Comment 2: *How were the parameters used in the Butler Volmer equations derived?*

Response: We thank the reviewer for the valuable suggestion to improve our manuscript. In this work, the parameters used in the Butler Volmer equations, including the exchange current density (i_0), the cathodic and anodic charge transfer coefficients (α_c , α_a) were empirical values preset by the similar simulation systems in a reasonable range, in consistent with those of the previous reports (Nature 2016, 537, 382; Energy Environ. Sci. 2024, 17, 510; J. Electrochem. Soc. 2023, 170 123502; ACS Sustainable Chem. Eng. 2017, 5, 4031; ACS Catal. 2023, 13, 916).

Comment 3: *In the discussion of the K^+ cation, there is no mention of how the electrical double layer is affected; the structure of this would certainly change the microenvironment near the catalyst as K^+ concentration changes. Can the authors discuss this, and comment on why this is relevant or not?*

Response: We thank the reviewer very much for the insightful and constructive suggestion to improve our manuscript. We are sorry that there is no mention of how the electric double layer is affected by K^+ cation in the previous version. Several results indicated that hydrated K^+ and H^+ showed competitive adsorption at outer Helmholtz plane (OHP) of the cathode in K^+ -containing acidic electrolyte (Nat. Catal. 2022, 5, 268; ACS Catalysis 2023, 13, 916; Adv. Energy Mater. 2022, 13, 2203603; Electrochem. Energy Rev. 2024, 7, 8). The electric fields generated by the cathode and by these adsorbed cations at OHP were in the same direction in the Stern layer while opposite in the diffusion layer. In K^+ -containing acidic electrolytes, due to the competitive adsorption of hydrated K^+ against H^+ at OHP, a chemically inert hydrated K^+ layer formed at OHP and shielded the electric field from the cathode in a long potential window. Thus, migration of H^+ was dramatically suppressed, which lowers the concentration of H^+ in the OHP and thus suppresses HER. Meanwhile hydrated K^+ strengthen the field in the Stern layer and stabilize key intermediates in CO_2 reduction (Supplementary Figure 18). The effect of this shielding electric field is enhanced with the increase of K^+ concentration. These analyses are consistent with that we observed.

In addition, according to the reviewer's the suggestion, the effective electric double layer capacitance (C_{dl}) of CD-Ag HPE in acidic electrolytes with different K^+ concentration were further analyzed in Fig. 5c and Supplementary Table 8. The C_{dl} of

CD-Ag HPE in acidic electrolyte was enhanced with the presence of K^+ . This enhancement effect showed an increasing trend when increasing the K^+ concentration. These results are consistent with the hypothesis, that is the hydrated K^+ physisorbed on the cathode in the acidic electrolyte modify the distribution of electric field in the double layer, which not only impedes HER by suppression of migration of H^+ , but also promotes CO_2 reduction by stabilization of key intermediates (Supplementary Fig. 18). The relevant results and discussions have been supplemented to the revised manuscript and Supplementary Information as follow:

“... In addition, the higher effective electric double layer capacitance (C_{dl}) of CD-Ag HPE in acidic electrolyte with high K^+ concentration, which was correlated with the electric field strength an enhanced electric field trend (Fig. 5c and Supplementary Table 8). These results are consistent with the hypothesis, that is the hydrated K^+ physisorbed on the cathode in the acidic electrolyte modify the distribution of electric field in the double layer, which not only impedes HER by suppression of migration of H^+ , but also promotes CO_2 reduction by stabilization of key intermediates (Supplementary Fig. 18)^{22,24,41}”.

“In K^+ -containing acidic electrolyte, due to the competitive adsorption of hydrated K^+ against H^+ at OHP, a chemically inert hydrated K^+ layer formed at OHP and shielded the electric field from the cathode in a long potential window^{14,15}. Thus, migration of H^+ was dramatically suppressed, which lowers the concentration of H^+ in the OHP and thus suppresses HER. Meanwhile hydrated K^+ strengthen the field in the Stern layer and stabilize key intermediates in CO_2 reduction. The effect of this shielding electric field is enhanced with the increase of K^+ concentration.”.

Supplementary Figure 18. Schematic diagram of electric double layer near cathode in **a**, $H_2SO_4 + KCl$ and **b**, H_2SO_4 electrolytes during CO_2RR . Outer Helmholtz Plane (OHP).

Fig. 5c. EIS Nyquist plots obtained in catholytes with different K^+ concentrations at pH 1. The inset in c shows the equivalent circuit.

Supplementary Table 8. EIS fitting parameters of different CO_2 -saturated catholytes by using same equivalent circuit.

Electrolyte	R_s (Ohm cm^2)	R_{ct} (Ohm cm^2)	C_{dl} ($\mu\text{F cm}^{-2}$)
0.05 M H_2SO_4 + 0 M KCl	8.9	2.7	587
0.05 M H_2SO_4 + 0.1 M KCl	5.9	1.8	631
0.05 M H_2SO_4 + 0.5 M KCl	2.0	1.3	711
0.05 M H_2SO_4 + 1.0 M KCl	1.1	1.1	726
0.05 M H_2SO_4 + 3.0 M KCl	0.5	0.9	905

Comment 4: Related to the above point, EIS measurements are presented as a function of K^+ concentration. Can information about the double layer be extracted based on this?

Response: We thank the reviewer very much for the insightful and constructive suggestion to improve our manuscript. According to the reviewer's suggestion, to evaluate the OHP generated by the hydrated K^+ layer, which is correlated with the electric field strength, the effective electric double layer capacitance (C_{dl}) of CD-Ag HPE in acidic catholytes with different K^+ concentration were obtained from the constant phase element (CPE) parameters and the two resistances (obtained from EIS by using the same equivalent circuit) using the Brug formula (Fig. 5c and Supplementary Table 8). The bare acid catholyte without K^+ showed a smaller C_{dl} value than that with K^+ cases. In addition, there was an evident increase of C_{dl} value for high K^+ concentration, indicating an enhanced electric field trend when increasing the K^+ concentration. These results are consistent with the hypothesis, that is the hydrated K^+ physisorbed on the cathode in the acidic electrolyte modify the distribution of electric field in the double layer, which not only impedes HER by suppression of migration of

H⁺, but also promotes CO₂ reduction by stabilization of key intermediates. The relevant results and discussions have been supplemented to the revised manuscript and Supplementary Information as follow:

“... In addition, the higher effective electric double layer capacitance (C_{dl}) of CD-Ag HPE in acidic electrolyte with high K⁺ concentration, which was correlated with the electric field strength an enhanced electric field trend (Fig. 5c and Supplementary Table 8). These results are consistent with the hypothesis, that is the hydrated K⁺ physisorbed on the cathode in the acidic electrolyte modify the distribution of electric field in the double layer, which not only impedes HER by suppression of migration of H⁺, but also promotes CO₂ reduction by stabilization of key intermediates (Supplementary Fig. 18)^{22,24,41}.”.

“In K⁺-containing acidic electrolyte, due to the competitive adsorption of hydrated K⁺ against H⁺ at OHP, a chemically inert hydrated K⁺ layer formed at OHP and shielded the electric field from the cathode in a long potential window^{14,15}. Thus, migration of H⁺ was dramatically suppressed, which lowers the concentration of H⁺ in the OHP and thus suppresses HER. Meanwhile hydrated K⁺ strengthen the field in the Stern layer and stabilize key intermediates in CO₂ reduction. The effect of this shielding electric field is enhanced with the increase of K⁺ concentration.”.

Comment 5: *Through the DFT simulations, would it be possible to construct a microkinetic model of the kinetics using the measured G's?*

Response: We highly appreciate the reviewer for the constructive suggestion to improve our manuscript. We agree that it is a very effective strategy to guide the experiment to construct a high-activity electrocatalytic CO₂ reduction reaction (CO₂RR) system by analyzing the kinetics under various potentials, current densities, K⁺ concentration, CO₂ coverage and pH within a microkinetic model. In general, it is feasible to construct a CO₂RR microkinetic model by combining a large number of DFT simulations (determine several key intermediates thermodynamic parameters of CO₂RR to CO on Ag catalysts, e.g. ΔG , ΔG_{TS} , the free energy change between the final and initial state, the kinetic barrier of transition state) with the assumption of several appropriate key parameters (e.g. concentration and coverage of species, reaction temperature, pressure).

However, in this work, the existing ΔG of *COOH and *H (Fig. 7) are insufficient to construct a relatively accurate microkinetic model of the kinetics due to the lack of ΔG_{TS} for several key intermediate species (e.g. *COOH, *CO and *H), and other several other key parameters (the concentration and coverage of key species, e.g. *COOH, *CO and *H). In the near future, we are trying to designing some experiments and simulation models that fully consider the unique structural characteristics of the hollow fiber electrode to confirm the concentration of reaction species and the key thermodynamic parameters, so as to build an optimized microkinetic model. This will greatly encourage us to further explore the reaction mechanism and screen out optimal reaction conditions for the construction of efficient CO₂RR systems.

Comment 6: *In the DFT simulations, how do the concentrations of K+ relate to*

concentrations of K⁺ in the electrolyte (e.g., in units of mM)?

Response: We thank the reviewer very much for the insightful and constructive comment to improve our manuscript. In this work, due to the limited solubility of KCl, we chose 3 M KCl as the highest K⁺ concentration acidic support electrolyte like many previous reports (Science 2021, 372, 1074; Nat. Commun. 2022, 13, 7596; ACS Catal. 2022, 12, 2357; Angew. Chem. Int. Ed. 2023, 62, e202309351). In fact, in a high K⁺ of 3 M KCl electrolyte, the ratio of K⁺ to H₂O is about 1/18. However, due to amplified electric field near the nanostructure and pore sites of cathode during the reaction, K⁺ could be enriched near the cathode, the local concentration of K⁺ at the surface of the cathode would be higher than that in the bulk (Nature 2016, 537, 382; Science 2021, 372, 1074; Nat. Commun. 2022; Angew. Chem. Int. Ed. 2023, 62, e202309351). Thus, in the DFT simulations, the ratios of K⁺ to H₂O in optimized model were set from 1/18 to 3/18 (Fig. 7), in consist with previous reports (Nat. Energy 2023, 8, 179; Nat. Energy 2022, 7, 978; Nat. Commun. 2022, 13, 7596).

Responses to Reviewer 4

Comment: *Shoujie Li et al. describe in their manuscript entitled “Ampere-level CO₂ electroreduction with single-pass conversion exceeding 85% in strong acid over a silver penetration electrode” very promising experimental results for the electrochemical CO₂ reduction employing a newly developed silver hollow fiber penetration electrode. Traditionally, alkaline electrolytes are used during the electrochemical reduction of CO₂ to CO over silver catalysts. These electrolytes lead to a strong reduction of the carbon efficiency due to the simultaneously occurring homogeneous reactions of CO₂ with hydroxide ions, typically resulting in a CO₂ utilization of approx. 50%. Therefore, it has been suggested to employ acidic electrolytes, which would enhance the carbon efficiency. On the other hand, the hydrogen evolution reaction is strongly favored at low pH values. For these reasons proper electrode designs and management of the local pH conditions at the silver catalyst are required. The presented results show very promising Faradaic efficiencies and single pass conversions, even at extremely high current density. Acidification of the KCl electrolyte with sulfuric acid towards a pH of around 1 led to the best performance and it could be clearly shown that the presence of potassium ions is required to achieve a high Faradaic efficiency. The experimental findings are also supported by simulations of the local electrolyte composition close to the electrode through a reaction-diffusion model as well as by DFT calculations. Overall, this is a very interesting and valuable contribution which can be published in Nature Communications after careful consideration of the following comments:*

Response: We highly appreciate the reviewer’s positive opinions. We also thank the reviewer very much for the careful reading and valuable suggestions to improve our manuscript. The manuscript and Supplementary Information have been revised according to these valuable suggestions.

Comment 1: *In the literature overview, I miss a recent publication by Löffelholz et al. (DOI: 10.1149/1945-7111/ad0eba) with quite similar trends of the FE towards CO, although over a very different electrode and at lower current density.*

Response: We highly appreciate the reviewer for the valuable comment to improve our manuscript. We agree with the reviewer that the recently published paper by Löffelholz et al. (J. Electrochem. Soc. 170 123502, DOI: 10.1149/1945-7111/ad0eba) showed quite similar trends of the FE towards CO at lower current densities, despite the use of a different electrode. In Löffelholz’s and our work, experimental results combined with a mathematical model showed that three distinct regimes depending on current density can be identified to account for the acidic electrocatalytic CO₂ reduction reaction (CO₂RR). The shift between these regimes strongly depends on current density and bulk electrolyte pH. In an acidic electrolyte, at low current densities (in the regime one), the local H⁺ concentration in the electrode is high, causing acidic hydrogen evolution reaction (HER) to dominate. When raising the current density (in the regime two), protons are consumed and the local pH increases. In this range, CO₂RR is the dominant

electrochemical reaction. These results are generally consistent with other reports (Science 2021, 372, 1074; ACS Catal. 2022, 12, 2357; Nat. Catal. 2022, 5, 564; Energy Environ. Sci. 2024, 17, 510).

However, further elevation of the current density ($>0.4 \text{ A/cm}^2$, in the regime three), the behavior across all bulk pH values converges toward the same characteristics, with the local pH being highly alkaline, while Löffelholz's CO₂RR becomes increasingly limited by CO₂ mass transfer of GDE, and alkaline HER becomes predominant. In contrast, in our work, we chose a silver hollow-fiber penetration electrode (HPE) as both the working electrode and the gas disperser comprising only single-component metallic silver with self-supported tubular structures and hierarchical porous walls. Such hollow-fiber configuration exhibited an excellent and reliable capability of enabling unlimited CO₂ supply to the three-phase sites by virtue of the oriented CO₂ flow penetration through the porous wall from the inside to the outside. The optimization of CO₂ mass transfer by unique HPE structure makes the efficient CO₂RR easily reach ampere-level current density (Nat. Commun. 2022, 13, 3080; Energy Environ. Sci. 2022, 15, 5391; Appl. Catal. B-Environ. 2023, 333, 122768; ACS Energy Lett. 2023, 8, 4867; Energy Environ. Sci. 2024, 17, 510). According to the comment, a brief conclusion and reference have been added in the revised manuscript as follows:

“...Thus, a higher value of j is required to consume a substantial amount of H⁺ and modify the surface pH to make it more favorable for the kinetics of CO₂RR. That is the shift of onset j for CO₂RR strongly depends on bulk electrolyte pH, which is consistent with the simulation results and other reports (Fig. 3a)²²⁻²⁶.”

In particular, the paper of “J. Electrochem. Soc. 170 123502, DOI: 10.1149/1945-7111/ad0eba” is cited as the reference 26.

Comment 2: *I recommend moving the details about the geometrical area of the hollow fiber electrode from the SI to the main manuscript, as it is very important for interpretation of the results. This would facilitate the understanding of the reader.*

Response: We thank the reviewer very much for the valuable suggestion to improve our manuscript. According to the suggestion, the details about the geometrical area of the hollow fiber electrode have been moved from Supplementary Information to the main manuscript as follows:

“...The CD-Ag HPE and Ag HPE both possessed the same exposed geometric area of 0.5 cm^2 ($S = \pi D_{\text{out}} L = 3.14 \times 400 \times 10^{-4} \times 4 = 0.5 \text{ cm}^2$, where S is the electrode area, D_{out} is the outer diameter of hollow fiber, and L is the length of hollow fiber). The CD-Ag HPE and Ag HPE both possessed the same exposed geometric area of 0.5 cm^2 ($S = \pi D_{\text{out}} L = 3.14 \times 400 \times 10^{-4} \times 4 = 0.5 \text{ cm}^2$, where S is the electrode area, D_{out} is the outer diameter of hollow fiber, and L is the length of hollow fiber). For the 10-tube CD-Ag HPE array electrode, the exposure geometric area was 5 cm^2 ($S = n \pi D_{\text{out}} L = 10 \times 3.14 \times 400 \times 10^{-4} \times 4 = 5 \text{ cm}^2$, where n is the number of hollow fiber tubes)”.

Comment 3: *The authors used KCl acidified with H₂SO₄ as electrolyte. However, the DFT calculations show no significant effect of Cl⁻ when compared to SO₄²⁻. This raises the question why KCl was used instead of K₂SO₄, introducing an anion species*

to the system that would possibly lead to undesirable anode reactions in a technical systems with mixed anolyte and catholyte.

Response: We thank the reviewer very much for the insightful and constructive comment to improve our manuscript. In this work, experimental results and density functional theory calculations suggested that the presence of K^+ in the acidic electrolyte shielded the electric field in the diffusion layer of the cathode and reduced the H^+ concentration around the active site, which not only suppressed the HER, but also stimulated CO_2 activation and conversion. And this effect was enhanced with the increase of K^+ concentration in the acidic electrolyte. These observations are consistent with many previous reports (Science 2021, 372, 1074; Nat. Catal. 2022, 5, 268; ACS Catalysis 2023, 13, 916; Nat. Commun. 2022, 13, 7596; Adv. Energy Mater. 2022, 13, 2203603).

Thus, on the one hand, to pursue a high acidic CO_2RR to CO activity by using acidic electrolyte with high concentration K^+ , we choosed KCl instead of K_2SO_4 as support catholyte like many other reported acidic systems (Science 2021, 372, 1074; Nat. Commun. 2022, 13, 7596; ACS Catal. 2022, 12, 2357; Nat. Synth. 2023, 2, 403; Angew. Chem. Int. Ed. 2023, 42, e202309351), because KCl (20°C, 1atm, ~4.6 mol/L) has a much higher solubility than K_2SO_4 (20°C, 1atm, ~0.64 mol/L) in aqueous solution. On the other hand, although the CO faraday efficiency in KCl and K_2SO_4 were basically the same at the high current density of 2 A/cm², the potential of 3 M KCl (-1.01 V vs RHE) was much lower than 0.5 M K_2SO_4 (-1.98 V vs RHE) due to the higher conductivity, which greatly affects their energy efficiency at the same high current density.

In addition, in this work, the cathodic and anodic compartments were separated by a Nafion 117 membrane, the catholyte and anolyte were cycled and refreshed at the same fixed flow rate of 50 mL min⁻¹, respectively. Thus, the mixing of catholyte and anolyte as well as formation of Cl_2 or HClO in anolyte was negligible, in consistence with the previous reports (J. Am. Chem. Soc. 2023, 1485, 8714; J. Am. Chem. Soc. 2021, 143, 3245; Angew. Chem. Int. Ed. 2022, 61, e202210432; Appl. Catal. B-Environ. 2024, 343, 123493; Science 2021, 372, 1074).

Comment 4: *The authors claim that the CO_2 loss by homogeneous reactions is negligibly small and justify this with experimental and simulation results. However, the experimental procedure for determination of the CO_2 loss should be described in greater detail. If I see it correctly, the gas-chromatographic method did not include the determination of carbon dioxide. Was the calculation based solely on the flow rate measurements? How precisely could the carbon balance be closed? I also recommend to use the more common term “carbon efficiency” instead of “ CO_2 loss” to represent results for CO_2 conversion during electrochemical vs. homogeneous reactions.*

Response: We thank the reviewer very much for the careful reading and insightful suggestion to improve our manuscript. We respond to these different points one by one as follows:

a) The authors claim that the CO_2 loss by homogeneous reactions is negligibly small and justify this with experimental and simulation results. However, the

experimental procedure for determination of the CO₂ loss should be described in greater detail. If I see it correctly, the gas-chromatographic method did not include the determination of carbon dioxide. Was the calculation based solely on the flow rate measurements? How precisely could the carbon balance be closed?

Response: We thank the reviewer very much for the insightful suggestion to improve our manuscript. We feel sorry that the experimental procedure for determination of the CO₂ carbonation (CO₂ loss in previous version) were not described in greater detail in the previous version. The reviewer is right that the gas chromatographic (GC) method in the previous version did not include the determination of CO₂. Actually, the calculation of CO₂ carbonation of different electrolytes with different pH values were based on the input and outlet of flow rate and CO, H₂ concentrations measurements. Specifically, the CO₂ carbonation and unreacted CO₂ were calculated as follows:

$$\text{SPCE} = \frac{\text{produced CO}}{\text{Input CO}_2} \times 100\%$$

$$\text{CO}_2 \text{ carbonation} = \frac{\text{Input CO}_2 - \text{Unreacted CO}_2 - \text{Converted CO}_2}{\text{Input CO}_2} \times 100\%$$

$$\text{Unreacted CO}_2 = V_{\text{outlet}} - \text{Prodeded CO} - \text{Prodeded H}_2$$

$$\text{Converted CO}_2 = \text{Prodeded CO}$$

Where the input CO₂ flow rate ($V_{\text{input CO}_2}$) of electrolysis cell was controlled by a Alicat[®] mass flow controller, and the outlet flow rate (V_{outlet}) of electrolysis cell was measured by an independent Alicat[®] mass flowmeter. The actual outlet gas of electrolysis cell contains unreacted CO₂, produced CO (from CO₂RR) and produced H₂ (from the HER). The produced CO and H₂ at given current density could be quantified by online GC.

In this work, the Alicat[®] mass flowmeter possessed a full scale of 50 sccm with the accuracy \pm (0.8% of Reading + 0.2% of Full Scale), as shown in the note **d** of Supplementary Table 3. In addition, the Alicat[®] mass flowmeter results were calibrated using a mixture (including CO, H₂, CO₂ and water vapour compositions at room temperature) that approximates the actual outlet gas composition (Supplementary Table 4). These results suggested the experiment errors caused by the Alicat[®] mass flowmeter could be negligible under the CO₂RR conditions. Furthermore, to verify our calculation of unreacted CO₂, the concentration of unreacted CO₂ in the outlet gas was measured by an independent online GC (Supplementary Figure 9). The results showed that the concentration of unreacted CO₂ obtained by chromatographic quantification was basically the same as that calculated by formula, and the maximum deviation was 2.5% (Supplementary Table 5). Thus, the carbon balance could be well closed within a reasonable measurement margin of error.

According to the suggestions, the detailed experimental procedures for determining the CO₂ carbonation have been supplemented in the revised manuscript and Supplementary Information as follows:

“The CO₂ SPCE, the CO₂ carbonation and unreacted CO₂ were calculated as follows:

$$\text{SPCE} = \frac{\text{produced CO}}{\text{Input CO}_2} \times 100\%$$

$$\text{CO}_2 \text{ carbonation} = \frac{\text{Input CO}_2 - \text{Unreacted CO}_2 - \text{Converted CO}_2}{\text{Input CO}_2} \times 100\%$$

$$\text{Unreacted CO}_2 = V_{\text{outlet}} - \text{Prodeded CO} - \text{Prodeded H}_2$$

$$\text{Converted CO}_2 = \text{Prodeded CO}$$

Where the input CO₂ (V_{input}) of electrolysis cell was controlled by a Alicat[®] mass flow controller, and the outlet flow rate (V_{outlet}) of electrolysis cell was measured by an independent Alicat[®] mass flowmeter. The actual outlet gas of electrolysis cell contains unreacted CO₂, produced CO (from CO₂RR) and produced H₂ (from the HER). The produced CO and H₂ at given current density could be quantified by online GC. Furthermore, to verify our calculation of unreacted CO₂, the concentration of unreacted CO₂ in the outlet gas was measured by an independent online GC (Supplementary Figure 9).”.

Supplementary Figure 9. The online GC curve from TCD over CD-Ag HPE at 2 A/cm² (operated in a CO₂-saturated 0.05 M H₂SO₄ + 3 M KCl catholyte and 0.05 M H₂SO₄ + 0.5 M K₂SO₄ anolyte). The concentrations of H₂, CO and CO₂ could be quantified in TCD Channel.

Supplementary Table 5. The Comparison between direct quantification by chromatography and calculation of unreacted CO₂ content of exhaust gas from electrolysis cell by formula method.

Conditions ^a	CO ₂ concentration ^b (%)	CO ₂ concentration ^c (%)	Deviation (%)
v _{CO2} = 30 sccm, j = 2 A/cm ²	74.8	74.4	0.5
v _{CO2} = 10 sccm, j = 2 A/cm ²	28.2	27.8	1.4
v _{CO2} = 5 sccm, j = 2 A/cm ²	7.8	8.0	2.5

^a The CO₂ electroreduction test was performed at pH = 1.

^b The CO₂ concentration was calculated by formula based on GC quantification of CO and H₂.

^c The CO₂ concentration was quantified by online GC.

b) I also recommend to use the more common term “carbon efficiency” instead of “CO₂ loss” to represent results for CO₂ conversion during electrochemical vs. homogeneous reactions.

Response: We highly appreciate the reviewer's constructive suggestion to improve our manuscript. We are sorry that there has been an inaccuracy of the previous description of the reduction of locally available CO₂ due to (bi)carbonate formation. According to the suggestion, we have used the more common term "CO₂ carbonation" instead of "CO₂ loss" to represent the reduction of local available CO₂ due to homogeneous reactions of (bi)carbonate formation.

Comment 5: The results of the stability test over 200 hours are very promising. However, silver is not stable in acidic environment and Löffelholz et al. (DOI: 10.1149/1945-7111/ad0eba) found a relatively rapid deterioration of their electrode performance in contact with acidified potassium sulfate electrolyte. Can the authors explain this difference?

Response: We highly appreciate the reviewer's positive opinions. We also thank the reviewer very much for the careful reading and valuable suggestions to improve our manuscript. According to the suggestion, we have studied the paper by Löffelholz carefully (J. Electrochem. Soc. 2023, 170, 123502, DOI: 10.1149/1945-7111/ad0eba). In Löffelholz's work, the authors reported that the reason for the deterioration of the stability of acidic systems was the degradation of Ag GDE in acidic systems. We hold the same opinion that the observed decline in performance could be attributed to the adhesives that support catalysts on carbon paper is easy to fall off in acidic environment, which cause the catalyst to gradually fall off from the catalyst layer and a rapid decline in performance. This is often observed even in neutral or alkaline GDE systems (ACS Energy Lett. 2019, 4, 3, 639; Energy Environ. Sci. 2020, 13, 977; ChemSusChem 2020, 13, 400; Chem. Eng. J. 2024, 482, 148944).

In contrast, in this work, the self-supported hollow fiber penetration electrode comprises only a single active composition with well-integrated porous structures without any additives and binders. This could ensure that the electrode structure remains stable during long-term testing, avoiding the deterioration of stability caused by the degradation and loss of catalyst. In addition, the simulation results showed that, at the presence of K⁺ in a strong acidic electrolyte (pH = 1), the local pH in contact with the electrode and electrolyte is always maintained at neutral or even alkaline under high current density (Fig. 2a). The result of inductively coupled plasma-optical emission spectroscopy (ICP-OES) of outlet electrolyte further showed no Ag dissolution was found during long-term test (Supplementary Table 10). Last but not least, in general, the metal activity of silver is behind hydrogen, which means that it is relatively stable in dilute acidic systems. To be noted that the metal activity of copper is ahead of silver, and many literatures have reported that Cu-based catalysts could stably and efficiently electrocatalytic CO₂ reduction in acidic systems (Science 2021, 372, 1074; Nat. Catal. 2022, 5, 564; Nat. Commun. 2022, 13, 7596; Energy Environ. Sci. 2024, 17, 510; Angew. Chem. Int. Ed. 2023, 62, e202309351).

Comment 6: The authors should describe in more detail how the iR correction was performed. At the current densities and gas flows through the fiber, intense bubbling can heavily impact the inner resistance of the electrolyte gap between WE and RE. How

was this taken into account? Moreover, why was an 85% *iR* correction chosen instead of the more common 90% or full correction?

Response: We highly appreciate the reviewer for the careful reading valuable suggestions to improve our manuscript. In this work, the 85% *iR* correction was applied by combining the solution resistance measured by EIS. The reviewer is right, the solution resistance values measured by EIS indeed affected by the application of current density and the CO₂ flows through the fiber. As shown in Supplementary Table 9, the *R_s* of different current densities fluctuated at 0.50 Ω cm², thus the average value of solution resistance obtained based on multiple measurements was used for *iR* correction.

In addition, a catholyte of 3 M KCl + 0.05 M H₂SO₄ with low resistivity was used in this work, which means a relatively low voltage drop over the catholyte. In order to avoid the overcorrected potentials, 85% *iR* correction was applied as many previous reports (J. Am. Chem. Soc. 2022, 144, 10446; Nat. Catal. 2019, 2, 251; Nat. Catal. 2019, 2, 1124; Science 2019, 364, 1091; Angew. Chem. Int. Ed. 2017, 56, 11394). The related discussions have been added in the revised manuscript and Supplementary Information as follows:

“All the applied potentials were recorded against the KCl-saturated Ag/AgCl reference electrode and then converted to those versus the reversible hydrogen electrode (RHE) with *iR* corrections using the following equation:

$$E(\text{vs. RHE}) = E(\text{vs. Ag/AgCl}) + 0.197V + 0.0591V \times \text{pH} + 0.85iR_s$$

Where *E* (vs. Ag/AgCl) is the applied potential, pH is the H⁺ concentration of the electrolyte solutions (Supplementary Table 6), *i* is the current density at each applied potential, and *R_s* is the average value of solution resistance obtained via multiple EIS measurements (Supplementary Tables 7, 8). In order to avoid the overcorrected potentials, 85% *iR* correction was applied as the previous reports^{17,48}.”

Supplementary Table 9. The solution resistances of CO₂-saturated 3 M KCl + 0.05 M H₂SO₄ catholytes at different *j*.

j (A/cm ²)	Solution resistance (Ohm cm ²)
OCV	0.52
0.1	0.54
0.5	0.50
1.0	0.47
1.5	0.51
2.0	0.48
2.5	0.53
3.0	0.49
4.0	0.47
5.0	0.52

Comment 7: In the experimental setup, a flat Pt-mesh is used as counter electrode. Could this result in an inhomogeneous distribution of potential in the working electrode between the side positioned towards the CE and the side facing away from it?

Response: We appreciate the reviewer for the careful reading and valuable comment to

improve our manuscript. The reviewer is right that an inhomogeneous distribution of potential in the working electrode between the side positioned towards the counter electrode (CE) and the side facing away from it. This is mainly due to the thickness of the working electrode (D_{out} of hollow fiber is 650 μm). According to the reviewer's comment, we added a group controlled experiment: The electrochemical experiment of the CO_2RR was conducted in three chambers electrolysis cell with a three-electrode system at room temperature, where the CD-Ag HPE was used as the working electrode, and two platinum meshes as CEs were placed on both sides of the working electrode, so that the both side of working electrode with a relative homogeneous distribution of potential. The result showed that under the same high current density, the faradaic efficiency of CO and H_2 in an electrolysis cell with two CEs is basically the same as that with single CE (Supplementary Figure). We speculated that in the experiment of single CE, due to the electrode thickness, although the inhomogeneous distribution of potential in the working electrode between the side facing the CE and the side facing away from it, since the D_{out} of hollow fiber is 650 μm , this is only 1/23 of the distance between the working electrode and the CE (about 1.5 cm). Therefore, the distribution of potential difference caused by electrode thickness has almost no effect on the CO_2RR performance.

Supplementary Figure. The schematic illustration of the electrolysis system with **a**, single counter electrode and **b**, two counter electrodes and corresponding **c**, CO, **b**, H_2 faradaic efficiencies of CD-Ag HPE in CO_2 -saturated 3 M KCl + 0.05 M H_2SO_4 catholytes at different current densities (pH 1, input CO_2 flow rate: 30 sccm).

Comment 8: Can the authors posit about the applicability of this type of electrode in large-scale systems?

Response: We highly appreciate the reviewer for the valuable comment to improve our manuscript. According to the comment, in order to demonstrate the scalability for practical applications using hollow fiber penetration electrodes, single-, 2-, 5- and 10-tube arrays of CD-Ag HPE (corresponding electrode geometric areas of 0.5, 1, 2.5 and 5 cm², respectively) were adopted and tested in a 2-electrode acidic system (pH = 1). The related experimental details, results and discussions have been added in the revised manuscript and Supplementary Information as follows:

“The eCO₂RR performance of the CD-Ag HPE array electrodes with different tube numbers in the 2-electrode acidic system (pH = 1) was performed using an ANS6050D (50 V, 50 A) direct-current source from ANS Power Co., Ltd. with constant current control applied.”.

“In order to demonstrate the scalability for practical applications using hollow fiber penetration electrodes, single-, 2-, 5- and 10-tube arrays of CD-Ag HPE were further adopted and tested in a 2-electrode acidic system (pH = 1). All CD-Ag HPE array electrodes with different tube numbers showed highly similar FE distributions of CO and H₂ at given high j range (Supplementary Figs. 32, 33). Thus, the j_{CO} over these CD-Ag HPE array electrodes also exhibited almost same rapidly growth trend with increasing j . Although CO FE and j_{CO} over the CD-Ag HPE array electrodes slightly decreased with increasing tube number and j , 10-tube CD-Ag HPE array still possessed over 90% of CO FE at high j of 4 A cm⁻². These results implied the potential scalability for practical applications using hollow fiber penetration electrodes.”.

Supplementary Figure 32. Optical images of **a**, 10-tube CD-Ag HPE array electrode. The gas-tight two-compartment electrolysis cell **b**, from a side view, and **c**, from a cross-section view during CO₂ electroreduction. The arrows show the directions of the CO₂ flow, exhaust flow, and electrolyte solution flow.

Supplementary Figure 33. The CO₂RR performance of **a**, single-tube CD-Ag HPE, **b**, 2-tube CD-Ag HPE array, **c**, 5-tube CD-Ag HPE array, and **d**, 10-tube CD-Ag HPE array at different j (0.5-4 A/cm²) in a 2-electrode system with CO₂-saturated 3 M KCl + 0.05 M H₂SO₄ catholytes. **e**, CO FE and j_{CO} over single-tube CD-Ag HPE, 2-tube CD-Ag HPE array, 5-tube CD-Ag HPE array and 10-tube CD-Ag HPE array in 3 M KCl + H₂SO₄ catholytes at different j .

Comment 9: In the manuscript and supplementary material, formatting is often inconsistent (e.g., $j_{CO,lim(gas)}$ vs. $j_{CO,lim(gas)}$). These inconsistencies should be fixed to improve the overall appeal of the publication. Additionally I recommend rephrasing of some unusual formulations like “pH is the *Pondus Hydrogenii* value”.

Response: We appreciate the reviewer for the careful reading and valuable suggestions to improve our manuscript. According to these suggestions, the format of text in manuscript and supplementary information have been carefully revised. In particular, the format of $j_{CO,lim(gas)}$ and $FE_{CO,lim(gas)}$ have been revised in the revised manuscript and Supplementary Information. In addition, we have replaced the words “pH is the *Pondus Hydrogenii* value of the electrolyte solutions” with “pH indicates the H⁺ concentrations of the electrolyte solutions” in the revised manuscript.

Comment 10: Page 6, line 6 from bottom: typing error “50 mm”.

Response: We thank the reviewer very much for the careful reading and constructive suggestions. We are very sorry for our previous typing error “50 mm”, and we have corrected it to “50 μ m” in the revised manuscript as follows: “The calculations based on reaction and diffusion of species model within a typical diffusion layer of 50 μ m showed that, at the presence of K⁺ in a strong acidic electrolyte (pH = 1), ...”.

REVIEWERS' COMMENTS

Reviewer #1 (Remarks to the Author):

The authors answered my objections.

One of my questions I had was "how do the authors avoid flooding the cathode". The paper answered that by noting that that are feeding CO₂ at such a high flowrate that they produce large numbers of CO₂ bubbles. Evidently the CO₂ bubbles are able to protect the cathode from the acid.

Overall this is a nice piece of work that deserves publication.

Reviewer #2 (Remarks to the Author):

Our comments and recommendations have been adressed with great detail and the revised manuscript may now be accepted for publication in Nature Communications.

Reviewer #3 (Remarks to the Author):

I am satisfied with the authors' changes to the manuscript. Thank you for all the work on this manuscript.

Reviewer #4 (Remarks to the Author):

The authors have very carefully revised their manuscript and addressed all critical points properly.

- An additional publication was included and thoroughly discussed.
- Important information was shifted from the SI to the main manuscript.
- Several experimental details were clarified.
- Additional experiments regarding the question of inhomogeneous potential distribution were conducted.
- A first concept for scale-up was presented and experimentally verified.

Overall, this is now a very interesting manuscript that could be published without further modifications.

848 **Responses to reviewers comments**

849 **Responses to Reviewer 1**

850 *Comment: The authors answered my objections.*

851

852 *One of my questions I had was "how do the authors avoid flooding the cathode". The*
853 *paper answered that by noting that that are feeding CO₂ at such a high flowrate that*
854 *they produce large numbers of CO₂ bubbles. Evidently the CO₂ bubbles are able to*
855 *protect the cathode from the acid.*

856

857 *Overall this is a nice piece of work that deserves publication.*

858 **Response:** We thank the reviewer very much for the positive comment.

859 **Responses to Reviewer 2**

860 *Comment: Our comments and recommendations have been adressed with great detail*
861 *and the revised manuscript may now be accepted for publication in Nature*
862 *Communications.*

863

864 *I co-reviewed this manuscript with one of the reviewers who provided the listed reports.*
865 *This is part of the Nature Communications initiative to facilitate training in peer review*
866 *and to provide appropriate recognition for Early Career Researchers who co-review*
867 *manuscripts.*

868 **Response:** We thank the reviewer very much for the positive comment.

869 **Responses to Reviewer 3**

870 *Comment: I am satisfied with the authors' changes to the manuscript. Thank you for*
871 *all the work on this manuscript.*

872 **Response:** We thank the reviewer very much for the positive comment.

873 **Responses to Reviewer 4**

874 *Comment: The authors have very carefully revised their manuscript and addressed all*
875 *critical points properly.*

876 *- An additional publication was included and thoroughly discussed.*

877 - *Important information was shifted from the SI to the main manuscript.*

878 - *Several experimental details were clarified.*

879 - *Additional experiments regarding the question of inhomogeneous potential*
880 *distribution were conducted.*

881 - *A first concept for scale-up was presented and experimentally verified.*

882 *Overall, this is now a very interesting manuscript that could be published without*
883 *further modifications.*

884 **Response:** We thank the reviewer very much for the positive comments.

885